

**Biomass-burning smoke properties and its interactions with marine stratocumulus clouds in WRF-CAM5 and southeastern Atlantic field campaigns**

Calvin Howes[1], Pablo E. Saide[1,2], Hugh Coe[3], Amie Nicole Dobracki[4], Steffen Freitag[5], Jim M. Haywood[6], Steven G. Howell[7], Siddhant Gupta[8], Janek Uin[8], Mary Kacarab[9], Chongai Kuang[8], L. Ruby Leung[10], Athanasios Nenes[11,12,9], Greg McFarquhar[13], Jens Redemann[13], Arthur J. Sedlacek[8], Kenneth L. Thornhill[14], Jenny P. S. Wong[9], Robert Wood[15], Huihui Wu[3], Yang Zhang[16], Jianhao Zhang[17,18], Paquita Zuidema[4]

[1]Dept. of Atmospheric and Oceanic Sciences, University of California, Los Angeles, 90064, USA

[2]Institute of the Environment and Sustainability, University of California—Los Angeles, Los Angeles, CA, USA

[3]Dept. of Earth and Environmental Science, University of Manchester, Manchester, M13 9PL, UK

[4]Rosenstiel School of Marine, Atmospheric, and Earth Science, University of Miami, Miami, 33149, USA

[5]State Agency for Nature, Environment, and Consumer Protection, Essen, North Rhine-Westphalia, Germany

[6]Dept. of Mathematics and Statistics, University of Exeter, Exeter, EX4 4PY, UK

[7]Dept. of Oceanography, University of Hawaii Manoa, Honolulu, 96822, USA

[8]Brookhaven National Laboratory, Upton, 11973, USA

[9]School of Earth and Atmospheric Sciences, Georgia Institute of Technology, Atlanta, 30332, USA

[10]Pacific Northwest National Laboratory, Richland, 99354, USA

[11]School of Architecture, Civil & Environmental Engineering, Swiss Federal Institute of Technology, Lausanne, Switzerland

[12]Institute of Chemical Engineering Sciences, Foundation for Research and Technology Hellas, Patras, GR26504, Greece

[13]School of Meteorology, University of Oklahoma, Norman, 73072, United States

[14]NASA Langley Research Center, Hampton, 23666, USA

[15]Dept. of Atmospheric Sciences, University of Washington, Seattle, 98195, USA

[16]Dept. of Civil and Environmental Engineering, Northeastern University, Boston, 02115, USA

[17]Chemical Sciences Laboratory, National Oceanic and Atmospheric Administration (NOAA), Boulder, 80305, USA

[18]Cooperative Institute for Research in Environmental Sciences (CIRES), University of Colorado, Boulder, 80305, USA

*Correspondence to:* Calvin Howes (calvinhowes@ucla.edu)



**Abstract**

A large part of the uncertainty in climate projections comes from poorly understood or constrained aerosol properties (e.g., particle size, composition, mixing state, aging processes) and aerosol-cloud interactions, as well as the difficulty in remotely sensing them. This is an issue especially in remote regions such as the southeast Atlantic, which exhibits large model spread due to the seasonal coexistence of extensive cloud cover and regional biomass burning smoke. Here we address these gaps by comparing

the WRF-CAM5 model to multi-campaign observations (ORACLES, CLARIFY, and LASIC) of the southeastern Atlantic region in August 2017 to evaluate a broad swath of the model's aerosol properties, processes, and transport, and the degree to which aerosol interactions with clouds are captured. Building on earlier work showing strong performance in model advection and mixing, we find that biomass-burning smoke aerosol size and composition are generally well-captured in the marine free troposphere,

except for a likely overprediction of dust in the accumulation mode (7-17% modeled dust fraction which is not present in the observations). Evaluating smoke aging trends, the model shows a steady increase in aerosol mean diameter and an unchanging composition as smoke ages, deviating from the observed trends that show a rise and subsequent fall in mean diameter over 4-12 days and a decreasing OA:BC ratio beyond 3 days. Both results are likely due to missing processes in the model that remove OA from the

particle phase such as photolysis and heterogeneous aerosol chemistry. The observed composition change from the free-troposphere to the marine boundary layer (MBL) is not fully captured in the model, especially the observed enhancement of sulfate from 11% to 37% by mean mass fraction in ORACLES, and from 11% to 26% in CLARIFY. This points to the importance of properly representing sulfate formation from marine dimethyl sulfide (DMS) emissions and in smoke-free parcels. Additionally, the

model does not capture the occurrence of an Aitken mode during clean and medium-smoke conditions in the boundary layer, likely pointing to misrepresentation of new particle formation. The model shows a persistent overprediction of aerosols in the MBL, especially for clean conditions, that multiple pieces of evidence link to weaker aerosol removal in the modeled MBL than reality. This evidence includes the model not representing observed shifts in the aerosol size distribution towards smaller sizes, the model

not capturing the relative concentrations of carbon monoxide compared to black carbon, model underprediction of heavy rain events, and little evidence of persistent biases in modeled entrainment. Average below-cloud aerosol activation fraction ($N_{CLD}/N_{AER}$) remains relatively constant in WRF-CAM5 between field campaigns (~0.65), while it decreases substantially in observations from ORACLES (~0.78) to CLARIFY (~0.5), which could be due to the model misrepresentation of clean aerosol

conditions. WRF-CAM5 also overshoots an observed upper limit on liquid cloud droplet concentration around $N_{CLD}$=400-500 cm$^{-3}$ observed in both ORACLES and CLARIFY and also overpredicts the spread in $N_{CLD}$. This could be related to the model often drastically overestimating the strength of boundary layer



vertical turbulence by up to a factor of 10 and having a bimodal—rather than the observed unimodal—probability distribution of updraft turbulent kinetic energy. We expect these results to motivate similar
evaluations of other modeling systems and promote model development in these critical areas to reduce uncertainties in climate simulations.

## 1.      Introduction

Among the anthropogenic radiative forcers quantified by the IPCC (International Panel on Climate Change), aerosols and their related cloud feedbacks have the largest uncertainty in global net
radiative forcing (Bellouin et al., 2020; Boucher et al., 2013; Myhre et al., 2013; Szopa et al., 2021). This is especially true of shallow stratocumulus clouds that top the boundary layer (Schneider et al., 2017).

Southern Africa is one of the largest regional sources of biomass-burning aerosols (BBAs) in the world, driven largely by human activities related to agricultural burning and land clearing annually during the dry season (Andela & van der Werf, 2014; Earl et al., 2015). Those emissions form large regional
plumes that, depending on meteorological conditions, advect westward and interact with the expansive, bright, semi-permanent stratocumulus cloud deck off the west coast (Adebiyi & Zuidema, 2016; Garstang et al., 1996; Kaufman et al., 2003; Miller et al., 2021; J. Zhang & Zuidema, 2021). The complexity of aerosols and cloud behavior makes the southeast Atlantic (SEA) a very large source of uncertainty in aerosol radiative effects (Redemann et al., 2021; Z. Zhang et al., 2016; Zuidema et al., 2016). These
radiative effects are a product of both the smoke plume properties and the underlying cloud albedo in the SEA (Bond et al., 2013; Chand et al., 2009; Cochrane et al., 2019; Eck et al., 2013; Kaufman et al., 2003; Leahy et al., 2007; Magi et al., 2008; Waquet et al., 2013).

This uncertainty may be understood and constrained by observations of aerosols and their interactions with clouds. Campaigns that utilize in situ observation platforms are critical to quantify aerosol-cloud
interactions and are less vulnerable to assumptions about aerosol properties or distribution than satellite measurements (Kaufman et al., 2003; C. Li et al., 2020). Different models generally utilize a wide range of parameter values for aerosol physical and chemical properties such as size distribution parameters, optical properties, hygroscopic water uptake, and density, among others (Che et al., 2021a; Gordon et al., 2018; Lu et al., 2018, 2021; Saide et al., 2020). Additionally, models will often include representation of
different aerosol aging and removal processes (Konovalov et al., 2019; Saide et al., 2012; Yu et al., 2019; Zawadowicz et al., 2020). The wide range of parameters and processes implemented plays a role in the uncertainties of their predictions, both of which can be constrained by field campaign data (Johnson et al., 2018).



A recent analysis examined multiple models' performance against observations from ORACLES

(ObseRvations of Aerosols above CLouds and their intEractionS). ORACLES was a NASA aircraft
campaign in 2016-2018 that studied biomass-burning smoke and clouds in the southeast Atlantic using
remote sensing and in situ instruments (Redemann et al., 2020). Compared to these observations in
September 2016 (Shinozuka et al., 2020), regional WRF-CAM5 was found to perform well among the
study cohort (vs. EAM-E3SM, GEOS-5, GEOS-Chem, and UK Unified Model [UM-UKCA], all global)

compared to smoke observations. WRF-CAM5 and GEOS-5 had finer horizontal resolution at ~30km,
UM-UKCA was 61km by 92km, EAM-E3SM was 100km, and GEOS-Chem was 2.5° by 2°. All were fed
by QFED2 fire emissions except UM-UKCA (FEER fires) and E3SM (GFED fires). All models' aerosol
schemes also contained the main fire emissions species of interest (black carbon, and organic aerosol),
along with other aerosols such as sea salt, sulfate, and dust. WRF-CAM5 had the smallest error in both

OA and BC concentration and spatial distribution, although OA still varied widely with a root-mean-
square error around 40% in the lower free troposphere (FT). Models in this study also consistently
exhibited biases towards a lower smoke layer base in the FT compared to lidar observations, and plume
top height differences of generally less than a model vertical grid cell. WRF-CAM5 was also found to
overestimate BC in the boundary layer offshore. CO was largely underestimated, especially in the lower

free troposphere and further offshore.

WRF-CAM5 was also compared to GEOS-5, CNRM-ALADIN, and UM-UKCA with a focus on
aerosol extensive and intensive properties important to the direct aerosol radiative effect (Doherty et al.,
2022). This study used model output overlapping all three ORACLES deployments, in September 2016,
August 2017, and October 2018. QFED2 emissions were used in both WRF-CAM5 and GEOS5, FEER

was used in UM-UKCA, and GFED in ALADIN. Doherty et al. (2022) found that WRF-CAM5 had a
bias towards low CO compared to observations in the core of the smoke plume (median CO bias -32% to
-13%). However, WRF-CAM5 outperformed GEOS5 and UM-UKCA in representing both BC and OA
concentrations at 1-3 km above the surface in 2017, which is the focus of this study, with a WRF-CAM5
median bias in BC concentration of -20% to +38%, and median bias in OA concentration -8% to +23% in

that year compared to observations. OA and BC in WRF-CAM5 were better represented in the 1-3 km
height range compared to GEOS5 in 2016 and 2018 as well, and the WRF-CAM5 bias was similar to or
lower than those of UM-UKCA in 2016 and 2017. The OA concentrations in the upper FT in both WRF-
CAM5 and GEOS5, especially between 4-6 km altitude, were 2-10 times higher than observations. BC
from 4-5 km was low in both models by a factor of 2. UM-UKCA showed biases of the same sign and

smaller magnitude for both OA and BC in the 4-6 km range. ALADIN biases of these quantities were not
reported. In summary, we expect that WRF-CAM5 captures the plausible ranges of major smoke



component concentrations in the year and altitudes studied here, where the largest smoke concentration and transport exist.

The first goal of this work is to analyze the performance of a fully online aerosol-resolving model, WRF-CAM5, in representing biomass-burning smoke processes. The model is compared to a wide range of observations from August 2017, when three field campaigns overlapped: ORACLES, CLARIFY-2017 (CLoud–Aerosol–Radiation Interaction and Forcing: Year 2017, Haywood et al., 2021), and LASIC (Layered Atlantic Smoke Interactions with Clouds, Zuidema et al., 2018). The second goal is to identify significant processes that may be missing or whose model representations cause substantial discrepancies between modeled and observed properties. Section 2 discusses the campaigns and data analyzed as well as the configuration of WRF-Chem, our sampling methods, and meaningful derived quantities. Section 3 compares observations with the model simulated smoke extensive properties such as number and mass concentrations, as well as intensive properties such as size, hygroscopicity, and composition in the free troposphere. We then address observations of changing smoke properties that suggest long-term aging, and that are not captured in the model. Simulated smoke in the marine boundary layer is also evaluated, especially utilizing observations from an ARM ground station. We further discuss aerosol composition, size distribution, and hygroscopicity and the representation of smoky and clean periods. Finally, we analyze model cloud activation and what it may reveal about underlying process biases.

## 2.    Methods

### 2.1    Observation systems

Model performance was evaluated by comparing model simulations with extensive in situ and remote sensing data from three field campaigns in the Southeastern Atlantic that coincided in August 2017—ORACLES, CLARIFY-2017, and LASIC. The model domain and field campaigns are shown in Fig. 1. The ORACLES campaign consisted of flights during the biomass-burning seasons in Southern Africa in 2016-2018 utilizing a mid-altitude P3 (2016-2018) and high-altitude ER2 (2016 only). The ORACLES base of operation was Walvis Bay, Namibia in 2016, and São Tomé Island, São Tomé, and Príncipe in 2017 and 2018. ORACLES flew various planned and opportunistic transects throughout the SEA (Redemann et al., 2021). The CLARIFY-2017 campaign in August-September 2017 flew an instrumented BAe146 FAAM aircraft from Ascension Island (ASI) in an approximately 5-degree radius around the island to sample smoke and clouds (Haywood et al., 2021). The LASIC campaign studied aerosol, clouds, and their radiation interactions using a highly-instrumented Atmospheric Radiation Measurement (ARM)



facility supported by the U.S. Department of Energy on Ascension Island from June 2016 to October 2017, covering two biomass-burning seasons (Zuidema et al., 2016; Zuidema, Sedlacek, et al., 2018). The data at ASI are supplemented by measurements from a permanent weather emplacement on the island, ~5 km away from the LASIC ARM station, operated by the UK Met Office. The selected instruments used in this analysis across all three campaigns are detailed in Table 1 and are described in detail in the campaign overview papers and references therein (Barrett et al., 2022; Dobracki et al., 2022; Haywood et al., 2021; Redemann et al., 2021; Taylor et al., 2020; Wu et al., 2020; Zuidema et al., 2018).



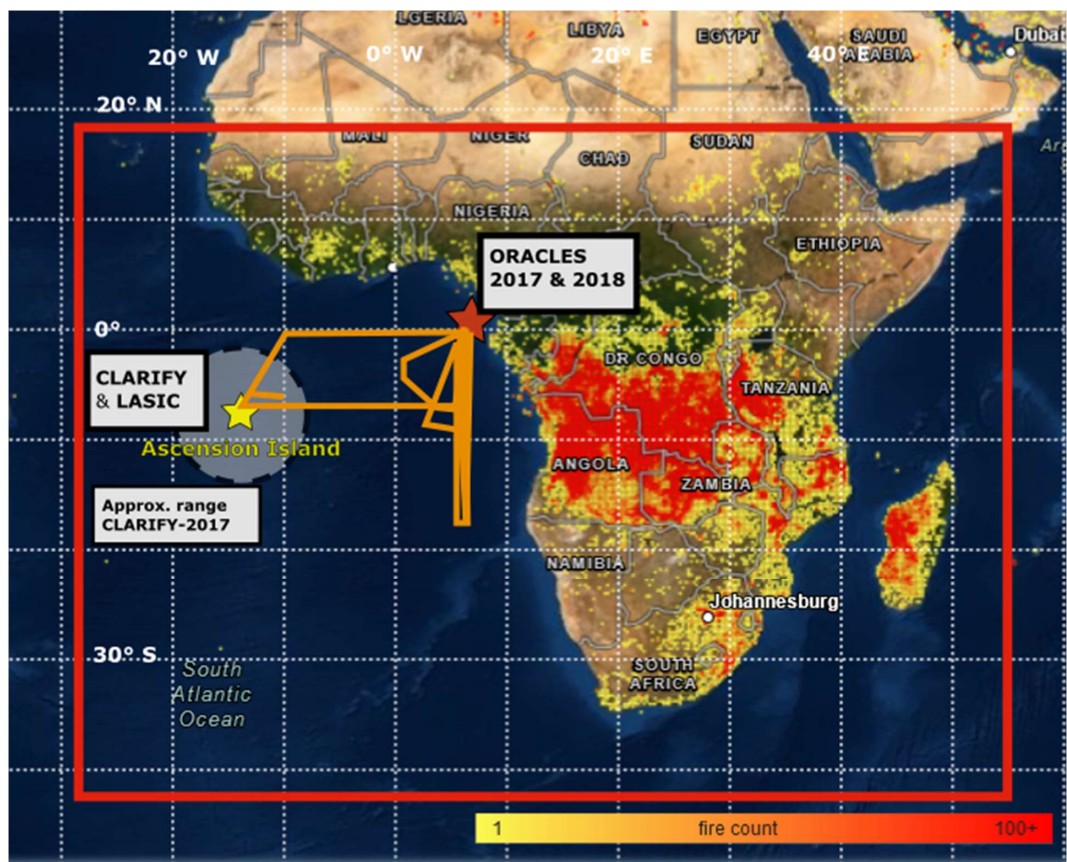

Figure 1: Domain of the WRF-CAM5 run for this study (red box) as well as the location of each observational campaign. Orange lines represent the approximate flight tracks of ORACLES 2017 flights. Color points are regridded fire detection counts in August 2017 from VIIRS/S-NPP and map layer obtained from NASA FIRMS.







**Table 1.** Summary of aerosol observations from field campaigns included in this study. Groups providing observations are noted in parenthesis and acronyms correspond to: DoE ARM – US Department of Energy Atmospheric Radiation Measurement; HiGEAR: Hawaii Group for Environmental Aerosol Research; UoM - University of Manchester; FAAM – Facility for Airborne Atmospheric Measurements; GIT – Nenes group at Georgia Institute of Technology; UND - Poellot group at University of North Dakota; BNL – Brookhaven National Lab; LRC - NASA Langley Research Center; UK Met O – UK Met Office. Instrument acronyms correspond to: AMS – High-resolution Time-of-Flight Aerosol Mass Spectrometer; ACSM – Aerosol Chemical Speciation Monitor; SP2 - Single Particle Soot Photometer; COMA – Carbon mOnoxide Measurement from Ames; VUV – NCAR Vacuum UV fluoromiter; UHSAS - Ultra-High Sensitivity Aerosol Spectrometer; LDMA – Long Differential Mobility Analyzer; SMPS - Scanning Mobility Particle Sizer; PCASP - Passive Cavity Aerosol Spectrometer Probe; CPC – Condensation Particle Counter; CCN - Cloud Condensation Nuclei; TAMMS – P3 Turbulent Air Motion Measurement System; AIMMS – Aircraft Integrated Meteorological Measurement System; CDP – Cloud Droplet Probe.

| Observable | ORACLES | CLARIFY / UK Met | LASIC (all instruments operated by DoE ARM) |
|---|---|---|---|
| Mass concentration (submicron, non-refractory) | AMS (HiGEAR) | AMS (UoM) | -- |
| Black carbon mass concentration | SP2 (BNL) | SP2 (UoM) | SP2 |
| Carbon monoxide | COMA (NASA Ames) | VUV (FAAM) | CO ANALYZERs |
| Aerosol size distribution | UHSAS, LDMA (HiGEAR) UHSAS (GIT), PCASP (UND) | PCASP (FAAM) | SMPS, UHSAS |
| Total aerosol number concentration | CPC (>3 nm and >10 nm) (HiGEAR) | PCASP (FAAM) | SMPS, CPC, UHSAS |
| Cloud Condensation Nuclei concentration | CCN (GIT); 0.1%, 0.2%, and 0.3% supersaturation | -- | CCN; 0.1%, 0.2%, 1.0% supersaturation |
| Aerosol hygroscopicity | CCN (GIT), AMS (HiGEAR) | AMS (UoM) | CCN; 0.1%, 0.2%, 1.0% supersaturation |
| Turbulence | TAMMS (NASA LRC) | AIMMS (UK Met O) | -- |
| Cloud droplet number concentration | CDP (UND) | CDP (FAAM) | -- |
| Ground-based Rain Accumulation | -- | Tipping bucket rain gauge (UK Met O) | RAIN non-tipping precipitation gauge |

## 2.2 Data processing

Here we outline specific methods of deriving key quantities from observations used to evaluate the model. Single-parameter hygroscopicity is estimated using two independent methods, both of which are widely adopted and described in Petters & Kredenweis (2007). First, we use Aerosol Mass Spectrometer (AMS) chemical mass and assumed density to calculate a simple volume-weighted average 200 hygroscopicity assuming internal mixing. We assume hygroscopicity values and density for each species in AMS/SP2 observations and the corresponding prescribed values in the model, as shown in Table 2. Second, we analyze the CCN concentration at 0.1%, 0.2%, and 0.3% in combination with the aerosol size



distribution to find the critical dry particle diameter of activation. For a given supersaturation (SS, the relative humidity above 100% where particles begin deliquescing) setting, the number-size distribution is

integrated from large bins down to small, and the diameter bin at which the integrated number concentration is first greater than or equal to the CCN concentration is the critical activation diameter $D_{crit}$. The diameter is used in the approximation formula $\kappa=(24/D_{crit})^3/(SS\%)^2$ (Petters & Kreidenweis, 2007). For ORACLES, we used the GIT UHSAS as it was configured to use the same aerosol sampling line as the CCN. For LASIC, we use the SMPS size distribution with the CCN at SS= 0.1%, 0.2%, and

1.0%. To the two UHSAS instruments in ORACLES (GIT and U. HI) and the single UHSAS in LASIC, we apply a size correction based on an observed bias towards under-sizing biomass-burning particles due to their large absorption (Howell et al., 2020).

Vertical turbulent kinetic energy (TKE) was calculated using vertical wind measurements from a high-resolution anemometer (Morales & Nenes, 2010). This calculation fitted a Gaussian curve to the

updraft spectrum, integrated over 1024 samples at 20Hz. The TKE ($m^2 s^{-2}$) was taken as $0.79*\sigma$, where $\sigma$ is the standard deviation of that Gaussian curve. This quantity is also output directly from WRF-CAM5, both selected in the vertical range of 100-700m that contained the majority of flat BL flight legs.

Inversion height in observations is calculated using two methods. First, the LASIC ARM value-added product included inversion heights and strengths derived by the Heffter method based on potential

temperature gradients (Pesenson, 2003). At ASI, this produced between 3 and 5 height values in each radiosonde dataset. We selected the primary capping inversion height as the one with the largest corresponding inversion strength. The inversion top in WRF-CAM5 was calculated as the local maxima of $\theta_{es}$ (effective potential temperature of a saturated parcel) below ~5 km, and within 1 km above the first layer with RH > 85% to denote the boundary layer, as well as the inversion base. We also applied the

same algorithm to the raw radiosonde profiles as applied to WRF-CAM5 to account for algorithm performance differences. The ARM data also included similar estimates of PBL depth from the algorithm of Liu and Liang (2010), but didn't report inversion strength so it is not used here. In all methods, inversion strength was calculated at each respective inversion height as a difference in potential temperature $\theta$ between inversion base and top.

## 2.3    Instrument intercomparison and selection

To make useful comparisons between models and observations from different field campaigns, we must understand the variability between instruments used in each campaign. To this end, Barrett et al. (2022) compared multiple cloud and aerosol instruments on ORACLES and CLARIFY aircraft as well as the LASIC ARM station and found broadly consistent measurements between similar instruments in each,





focusing especially on the joint flight day (18 August 2017) on which both the ORACLES and CLARIFY
aircraft flew close together through smoke and clouds near Ascension Island. This comparison showed
there was good agreement for BC, aerosol number concentration, and aerosol size distributions. Chemical
compositions from the SP2 and ToF-AMS were also shown by Barrett et al. to be within instrument
uncertainty, and within one standard deviation between ORACLES and CLARIFY for most species. The

ORACLES AMS reported a 40% higher sulfate mass that was not attributable to likely instrument
uncertainty or postprocessing. The LASIC ACSM also measured composition, but resulting OA and SO$_4$
measurements showed a tendency towards 2-4x lower mass concentrations than either the ORACLES or
CLARIFY AMS. Diagnosing the reason for this difference is beyond the scope of this work. For the sake
of consistent comparison between instruments without confounding uncertainty, we will focus on the two

aircraft-mounted AMS instruments that have been shown to perform similarly.

Additionally, we performed a volume closure assessment between ORACLES mass (AMS) and
aerosol size (U. Hawaii UHSAS and PCASP) instruments for measurements in the free-troposphere.
WRF-CAM5 prescribes aerosol density per species as shown in Table 2, and we assumed values as
shown for AMS-measured species. We found well-correlated volume closure with low error between the

UHSAS, PCASP and AMS (Fig. A1). This suggests first that the PCASP, with its higher upper size range
around 3 μm, was not capturing aerosols that would have been missed with the UHSAS upper size cutoff
of 1 μm. Second, both correlated well with the AMS total volume, given the density assumptions below.
This tells us that there was not significant aerosol mass beyond what the AMS was able to capture, such
as dust and sea salt. This is also evident in the UHSAS size distributions (see section 3.1.1).

**Table 2**. Assumed density and hygroscopicity of aerosol species. In WRF, values are prescribed and used in
volume calculations. In AMS, values are taken from literature (Jimenez et al., 2009; Shinozuka et al., 2020; Wu et
al., 2020).

| | POA | SOA | BC | SO4 | NH4 | NO3 | Chl | Dust |
|---|---|---|---|---|---|---|---|---|
| WRF-CAM5 ρ | 1.00 g/cm³ | 1.00 g/cm³ | 1.70 g/cm³ | 1.77 g/cm³ | N/A | N/A | 2.60 g/cm³ | 1.90 g/cm³ |
| Obs ρ | 1.27 g/cm³ | N/A | 1.77 g/cm³ | 1.77 g/cm³ | 1.77 g/cm³ | 1.77 g/cm³ | N/A | N/A |
| WRF-CAM5 κ | 0.10 | 0.14 | 1.00E-10 | 0.507 | N/A | N/A | 1.16 | 0.068 |
| Obs κ | 0.10 | N/A | 1.00E-10 | 0.507 | 0.5 | 0.5 | 1.16 | N/A |

Chloride mass concentration is not used from the ORACLES AMS data as it provided
unrealistically high values in the mid and upper free troposphere. This is consistent with the processing of

the public data from the LASIC ACSM and CLARIFY AMS, which have similar issues measuring



chloride in biomass smoke. As mentioned above, a volume closure suggests that there is very little chloride by mass in the free troposphere, so we expect little impact on FT smoke properties.

The CLARIFY CCN is not analyzed for this work, as our primary usage of CCN data is to calculate hygroscopicity. PCASP, as the available instrument resolving size distributions in the

CLARIFY dataset, has both a lower size resolution and a larger lower-end size cutoff (~100 nm) than the UHSAS that both lead to large uncertainty in deriving κ.

## 2.4    WRF-Chem configuration

This work uses Weather Research and Forecasting with Chemistry (WRF-Chem) model, version 3.4. We utilize the Community Atmosphere Model (CAM5) aerosol and physics parameterizations (Y. Chen

et al., 2015; Ma et al., 2014; Y. Zhang et al., 2015) which include the Modal Aerosol Module (MAM3) aerosol representation with 3 lognormal size modes (X. Liu et al., 2012), Fountoukis and Nenes (FN) series cloud droplet activation (Fountoukis & Nenes, 2005), ice nucleation via Niemand et al. (Niemand et al., 2012), and Bretherton-Park (UW) boundary layer turbulence scheme (Bretherton & Park, 2009). The aerosol scheme is coupled with gas-phase chemistry of the Carbon Bond Mechanism version Z

(CBMZ) (Zaveri & Peters, 1999). Natural dust emissions come from the "DustDEAD" emissions algorithm (Zender et al., 2003). This configuration of WRF-CAM5 is used because it resembles the configuration used in global climate models, improvement of which is an extended goal of this research. We also use this model because it contains chemistry, aerosol-cloud feedbacks, and aerosol-radiation feedbacks which are highly relevant for absorbing smoke and aerosol-cloud interactions. The model was

configured with a horizontal grid resolution of 36 km with 72 vertical layers at 5hPa spacing, and a domain covering the southern burning region of Africa and the southeastern Atlantic. The National Centers for Environment Prediction-Final (NCEP-FNL) climatology (National Centers for Environmental Prediction, National Weather Service, NOAA, U.S. Department of Commerce, 2000) is used to initialize meteorology and boundary conditions. The anthropogenic emissions and trace gases for this study come

from EDGAR-HTAP (Janssens-Maenhout et al., 2012), while fire emissions come from QFED2 (Darmenov & da Silva, 2015). QFED2 is provided at daily time resolution and 0.1° spatial resolution. A superimposed diurnal cycle is applied to resemble real burning trends, such as that applied to an NCAR WRF-Chem build in Ye et al. (2021).

As described in previous work (Diamond et al., 2022), there is no subgrid shallow cumulus scheme

enabled as we discovered that it led to significant suppression of the boundary layer height and clouds compared to observations. Also, we use no subgrid scheme for smoke plume injection, and emissions are placed within the 1st model level. This is done as fires in the region tend to be small and the boundary



layers over land are deep, so few injections above the boundary layer are expected. This assumption produces reasonable smoke layer heights over the southeast Atlantic (Shinozuka et al., 2020). MAM3 uses 3 predefined lognormal size modes with fixed width and mean diameter at emission, after which the mass and number evolve freely but the width is kept fixed. We also changed emissions to exclude the "other PM2.5" category (i.e., total PM2.5 - OC - BC) in the emissions files. Before our change, this was then added to the accumulation mode aerosol mass in the dust category. With "other PM2.5" classed as dust, the modeled dust concentration in the LFT was ~8 μg m$^{-3}$ across ORACLES samples and ~5.5 μg m$^{-3}$ across CLARIFY samples, or about 30% and 35% of the total accumulation mode mass in those samples respectively. We consider this dust mass to be unrealistically large mass when comparing it to observations of low-dust conditions in the FT during ORACLES and CLARIFY. Cloud droplets are activated in the model based on both aerosols at cloud base and further secondary aerosol activation within the cloud.

Following suggestions in recent work (Diamond et al., 2022; Shinozuka et al., 2020) comparing multiple models to ORACLES data, as well as our own calculations in the free troposphere, we adjusted aerosol size parameters of the accumulation mode—applying across all species—to bring the model closer in line with observations. In particular, the geometric mean diameter (i.e., count mean diameter) of the accumulation mode emissions was changed from 110 nm to 150 nm and its standard deviation was changed from 1.8 to 1.5. These changes are consistent with both ORACLES observations and estimates in literature of crop-burning primary emission sizes (Hays et al., 2005; X. Li et al., 2007; Winijkul et al., 2015; H. Zhang et al., 2011). The refractive index of organic carbon is set at $1.45 + 0i$, and that of black carbon is $1.85 + 0.71i$ for optical property calculations.

The model is reinitialized every five days and runs for seven days at a time, with the first two days used to spin-up the meteorology. The aerosol conditions are carried over from day 5 of the previous seven-day run cycle and the meteorology is reinitialized to NCEP-FNL. This allows aerosols to evolve continuously while meteorology remains relatively close to reanalysis. This setup also allows several days for aerosol-climate feedbacks to manifest, such as smoke heating in the free troposphere, which may substantially alter subsidence and transport (Adebiyi & Zuidema, 2016).

We also uncovered a bug in the diagnostic CCN number calculations within the mixing and activation scheme: the model was not calculating a dynamic mean aerosol diameter based on total mass and number per mode, but instead was using a prescribed value from the MAM aerosol mode definitions. This led to an overestimation of all CCN concentrations in the output, although cloud activation was unaffected as CCN is recalculated separately based on the dynamic particle diameter. This bug was reported to the WRF-Chem development team, who have now released a fix. However, any WRF-Chem builds up to



v4.2.1 or model source code obtained before January 15th, 2021 may be affected. This bug may have substantially impacted studies using WRF-Chem that reported on CCN concentrations directly, a not-uncommon practice when reporting on aerosol-cloud interactions. Note that further usage of the term "WRF" or "WRF-CAM5" in this work refers exclusively to the configuration described here.

## 2.5      Analytical Methods

In the free troposphere, our goal was to select smoky periods during relatively level flight legs. We focus on periods of uniform smoke behavior in the free troposphere in particular to eliminate background aerosol signals and reduce in-sample variability. We therefore selected 8-minute segments from 1-minute-merged data that contiguously met the threshold criteria for altitude and smokiness. This 8-minute time interval represents roughly 55-100 km of aircraft travel, which in a straight line would pass through 2 model grid cells on average and was chosen to smooth the observational variability. In ORACLES, we selected data for aircraft height > 1200m, RH < 80%, and CO concentration >120 ppb. We also limited samples to those segments with average total aerosol mass concentrations > 5 $\mu g/m^3$ and BC > 100 $ng/m^3$. This is similar to the Shinozuka et al. (2020) threshold of BC > 100 $ng/m^3$ to identify smoke plumes and we incorporate AMS data availability as a key requirement for our analysis. In CLARIFY, we selected for the same height and RH, CO > 100 ppb, total aerosol mass > 1 $\mu g/m^3$, and BC > 50 $ng/m^3$ to account for further plume dispersion over long distances. In both campaigns we selected flight legs with minimal altitude changes (less than 100m over the sample period) to avoid sampling vertically-stratified distinct smoke layers. We then extracted comparable observations and colocated model quantities for each variable of interest.

We treat the marine boundary layer as generally well-mixed for the purposes of smoke comparison. Boundary layer segments were selected in ORACLES by a threshold of altitude Z < 1000m, RH < 95%, and BC concentration > 100 $ng/m^3$. Boundary layer segments were selected in CLARIFY by z < 1200 m, RH < 95%, and CO > 100 ppb. These thresholds were used to maximize data availability and consistency and avoid sampling within clouds. The higher altitude threshold in CLARIFY is to allow more data samples with the typically deeper and decoupled boundary layer near ASI, and the usage of a CO threshold rather than BC for smokiness in CLARIFY is a compromise considering data availability from the SP2.

A different modeling system was used to estimate smoke age, using the WRF Aerosol Aware Microphysics (WRF-AAM) configuration that was used regularly and reliably to forecast smoke transport throughout the ORACLES campaign (Redemann et al., 2021) and as such, we expect it to provide a reasonable estimate of the observed smoke age. To estimate smoke age, biomass burning tracers tracking



each day of emissions over the whole African continent were added to WRF-AAM. The concentration of the tracer from each day was used to calculate a weighted average of the emission day at a given point in space and time, thus giving an estimate for the average age of that plume. The age extracted from WRF-AAM is used as an age estimate for WRF-CAM5 and the observations. Given differences in transport between all three of WRF-AAM, WRF-CAM5, and reality, the WRF-AAM age estimation method does not provide a perfectly Lagrangian age estimate following the plume itself. However, it still gives insight into bulk property changes in the smoke over time.

Clouds are analyzed by comparing the vertical profile of droplet number concentration (CDNC or $N_C$) to below-cloud aerosol concentration. Cloud droplet data points are based on averaging 1-second resolution data as the P-3 and Bae146 FAAM aircraft profiled a cloud layer. These passes occurred over a relatively short horizontal distance (approx. 3 km) relative to the size of stratocumulus cloud decks, thus they are treated as vertical cloud profiles. When sawtooths were flown (diving up and down through a cloud layer multiple times in close succession), the profile-mean values from each single cloud profile were then averaged together. Selecting cloud profiles in ORACLES followed the same criteria as Gupta et al., (2021) and CLARIFY cloud selection used similar methods. Following the methods of Diamond et al. (2018), we report droplet-mass-weighted $N_C$ recorded by the same probe. This de-emphasizes regions of extremely thin clouds and emphasizes regions with high liquid water.

For WRF, we calculate below-cloud aerosol by averaging across the two grid cells immediately below the cloud base, which were defined by a weighted droplet concentration threshold of 0.1 cm$^{-3}$. For observations, the below-cloud aerosol was calculated as an average over the roughly 100 m sampled below the cloud base. To account for differences in vertical placement of clouds and marine boundary layer heights in the model vs. observations, all model cells below 3 km with weighted $N_C$ above the 0.1 cm$^{-3}$ threshold were considered regardless of vertical structure. The model grid cells were co-located using the average latitude and longitude of the transect.

## 3.        Results

### 3.1 Free troposphere smoke

#### 3.1.1        Physical properties

The free troposphere is the logical starting point to evaluate model representation of biomass-burning smoke aerosols. The smoke from the continent travels throughout most of the southeast Atlantic (SEA) region in the free troposphere, with occasional entrainment into the boundary layer (Diamond et al.,



2018). As a result, the lower free troposphere (cloud-top up to roughly 3km) has a much higher and more consistent concentration of smoke than the boundary layer. Additionally, the boundary layer is itself a source of new aerosol particles that confound the smoke signal—primarily sulfates, salts, and organic particles from sea spray (Meskhidze et al., 2013; Zorn et al., 2008). The capping inversion frequently keeps this aerosol population from mixing heavily into the free troposphere, and so it can constitute a large fraction of the aerosol mass in smoky conditions.

Our analytical framework here supports and expands earlier conclusions about WRF-CAM5 performance. We find that the model FT accumulation-mode mean number concentration is biased high by 28% compared to ORACLES observations (Fig. 2a) and by 38% compared to CLARIFY (Fig. 2b). WRF-CAM5 volume concentration is comparable to ORACLES (Fig. 2c, WRF-CAM5 mean bias=+36% vs. UHSAS, -16% vs. PCASP) and relatively high compared to CLARIFY (Fig. 2d, WRF-CAM5 mean bias=+111% vs. PCASP). Total aerosol mass concentration simulated by WRF-CAM5 has a mean bias of -10% compared to ORACLES and +108% compared to CLARIFY (Fig. 2e-f), tracking the trend in volume. These larger relative discrepancies with CLARIFY may be explained by a lack of mass loss through aging in WRF-CAM5 or insufficient scavenging, which will be discussed later.

WRF-CAM5 represents the range of geometric mean diameters well and is closest to the U. Hawaii UHSAS (Fig. 3a). The 25th-75th percentiles of samples of geometric mean diameter are as follows: WRF, 186-208 nm; UHSAS, 176-196 nm; PCASP, 220-244 nm; LDMA, 208-231 nm. The model lognormal distribution also closely follows the spread and mean of observations on a representative sampling day (24 Aug 2017), despite a bias towards high model number (Figs. 3b-c). The variability between instruments is not unexpected and we conclude that, after observationally-constraining smoke aerosol size at the point of emission, WRF-CAM5 can successfully represent the mean particle diameters after transport to the SEA to within instrument uncertainty.



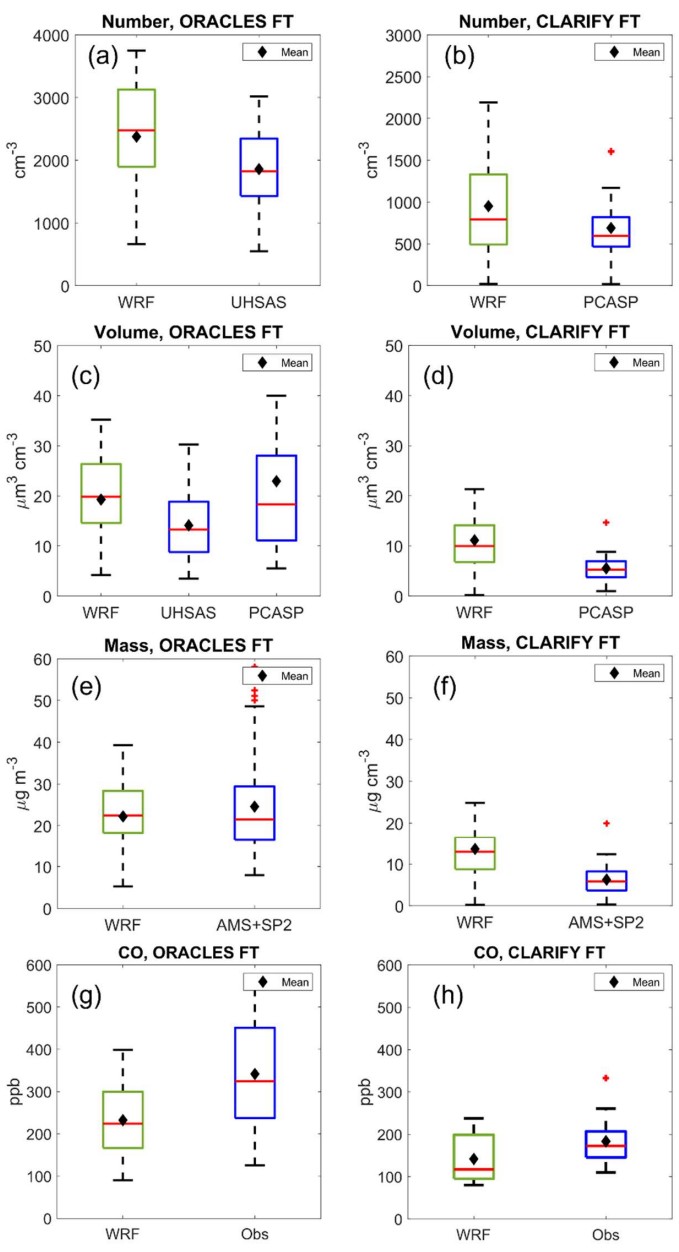

Figure 2: Extensive properties of smoke in the free troposphere (FT), comparing WRF-CAM5 and appropriate instruments from both ORACLES and CLARIFY in 2017. Red line represents sample median, and black diamond represents mean. a-b) Number concentration; c-d) Volume concentration; e-f) Mass concentration, compared to combined AMS and SP2 mass measurements; g-h) CO concentration.



Two other important features are visible in the number and volume distributions of free-tropospheric smoke from ORACLES. In the number size distributions (Fig. 3b), there is a dominant accumulation mode and extremely small number concentration of coarse mode (>1 μm) or Aitken mode (<40 nm)

particles. This holds across >90% of smoky ORACLES samples in the FT on other days (not shown). The lack of coarse mode is supported by the volume size distribution from PCASP, (Fig. 3c, green) showing that in the great majority (~95%) of ORACLES cases there is not a substantial volume of coarse particles such as mineral dust or sea spray, at least below ~3 μm. The volume closure between the AMS, PCASP, and UHSAS supports this. The smoke sampled here is days old, and any new particle formation that would generate an Aitken mode was likely in the past near the source in Africa. The LDMA, with its

lower size range of around 10 nm, supports this notion.

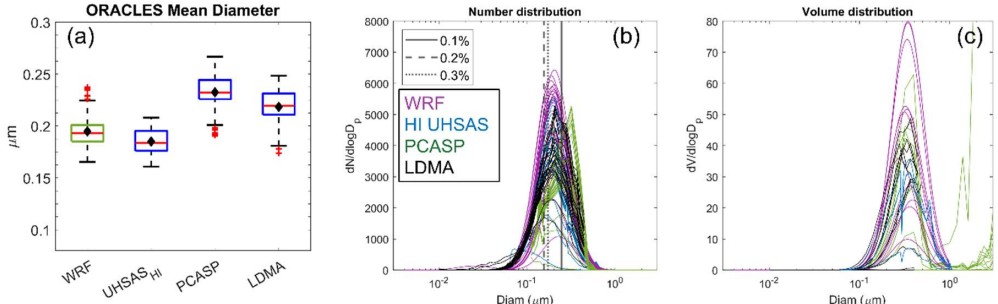

Figure 3: Size properties in the free troposphere, from both WRF-CAM5 and ORACLES instruments. CLARIFY data is excluded here for lack of available instruments with a comparable size range. a) geometric mean diameter across all FT samples deemed smoky and flat enough. b-c) Number and volume size distributions of same instruments from 31 Aug 2017, showing both WRF-CAM5 nucleation and accumulation mode. Superimposed on fig. 3b are the calculated $D_{crit}$ based on the CCN and GIT UHSAS at the three primary supersaturation settings.

The average composition fractions across the FT samples in ORACLES and CLARIFY are shown in Fig. 4. Of note, the AMS instruments were not able to accurately measure chloride salts or detect nonrefractive mineral dust, which the model includes in its hygroscopicity calculations. As noted above

these likely are a minimal mass component in observations. WRF-CAM5 also lacks aerosol nitrate and ammonia in its implementation of MAM3. WRF-CAM5 also treats aerosol modes as internally mixed, similar to calculations based on the AMS. μ

The single-parameter hygroscopicity factor κ is biased low in the WRF-CAM5 FT compared to both AMS+SP2 and CCN-based ORACLES FT estimates, although it is closest to the AMS+SP2 in median

and spread (WRF-CAM5 median: 0.125; AMS+SP2 median: 0.172; CCN+UHSAS median: 0.185) (Fig. 4a). Performance is similar against CLARIFY (not pictured), with median CLARIFY WRF-CAM5 κ=0.14 and AMS+SP2=0.20. In the ORACLES boundary layer, the median hygroscopicity is as follows: WRF-CAM5 (all aerosol species) = 0.19; WRF-CAM5 (excluding chloride to match the AMS) = 0.14;



AMS+SP2 = 0.27. Through the CLARIFY BL, the median κ is as follows: WRF-CAM5 (all aerosol
species) = .17; WRF-CAM5 (excluding chloride to match the AMS) = .14; AMS+SP2 = 0.21. The
increase in AMS-based κ in the BL tracks with the elevated sulfate in the BL compared to the FT across
both campaigns. The CCN and UHSAS from ORACLES had irregular availability and discontinuous
SS% sampling in the BL compared to the FT and are unable to be separated by SS% as done in the FT.
Thus, MBL κ calculations based on CCN are not included in this comparison.


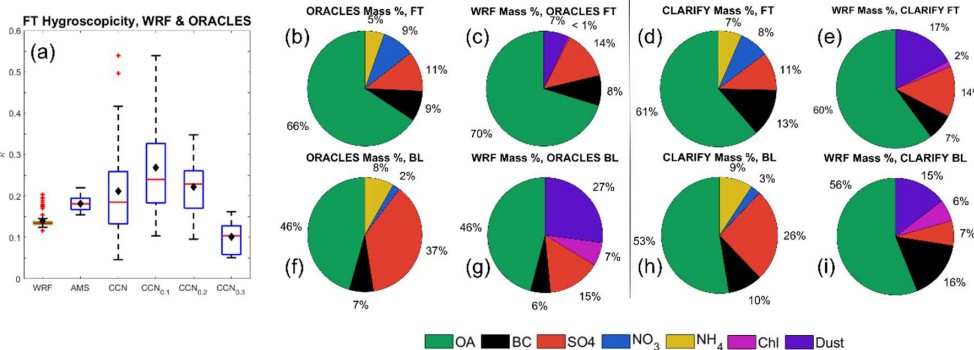

Figure 4: Chemical and hygroscopic properties of smoke in the ORACLES FT+BL, and composition in the
CLARIFY FT+BL. a) Hygroscopicity from WRF-CAM5 and data from the Nenes group, grouped and then
disambiguated by CCN supersaturation setting. Calculations were made with the CCN and GIT UHSAS together;
b-e) average composition by mass fraction of smoke in ORACLES and CLARIFY FT, and colocated WRF-CAM5
samples; f-i) average composition by mass fraction of smoke in ORACLES and CLARIFY BL, and colocated
WRF-CAM5 samples. Model OA here includes secondary OA, a distinct model variable. WRF-CAM5 SOA was
generally less than 3% of total mass.

We suggest a few potential explanations for the low model κ bias. First, in our configuration WRF-
CAM5 lacks nitrate or ammonia aerosols, both of which increase the bulk hygroscopicity since $\kappa_{NO3}$ and
$\kappa_{NH4}$ are both roughly assumed to be 0.5. Second, WRF-CAM5 retains around 10% of total aerosol mass
as dust, which in the model has very low hygroscopicity at 0.068. This dust comes from the natural dust
emission scheme and is not related to fire emissions. Third, the prescribed properties for OA in the model
may not be physically accurate. WRF-CAM5 uses a prescribed $\kappa_{OA}$=0.1 and density of 1.0 g/cm$^3$. The set
density of 1.0 g/cm$^3$ for OA in WRF-CAM5 is low compared to both lab studies (Kuwata et al., 2011) and
campaign-wide assumptions used in other studies, such as 1.27 g/cm$^3$ (Wu et al., 2020). An erroneously
low model density leads to a larger volume, which decreases κ since it is a volume-weighted mass
average. An OA density of 1.27 g/cm$^3$ also produces the best volume agreement between the ORACLES
AMS, UHSAS, and PCASP. Existing literature measuring the density of biomass-burning aerosol (BBA)
organics over long aging periods is generally limited, but there is evidence that OA density is increased
by at least 30%--and up to 90%--over the course of a few days (Dinar et al., 2006; Kuwata et al., 2011).



$\kappa_{OA}$ may realistically have values ranging from 0 to 0.2, with nonlinear dependence on age and oxidation

level (Duplissy et al., 2011; Kacarab et al., 2020; Kuang et al., 2020; Wonaschütz et al., 2013).

WRF-CAM5 and AMS show a similarly narrow range in $\kappa$, despite the bias in mean. This indicates that the average bulk composition fractions of observed BBAs vary little, as far as the AMS is capable of measuring. The hygroscopicity based on CCN shows a notably large spread, however. This is partially a result of convoluted instrument uncertainties (combining CCN and UHSAS instrument variability) and

partially a result of the $\kappa$ estimation strategy. The AMS measures bulk chemical mass while the $\kappa$ based on UHSAS + CCN critical diameter ($D_{crit}$) depends upon the properties of the aerosol population around that size. At 0.1% CCN SS, $D_{crit}$ fell in the range of 100-250 nm, near the middle of the accumulation mode in most cases. At 0.2% and 0.3%, $D_{crit}$ was in the range of 60-180 nm, with $D_{crit}$ at 0.3% ~10nm lower on average than at 0.2%. Values of $\kappa$ tend to be higher at 0.1% SS (mean $\kappa$=0.27) than at 0.2%

(mean $\kappa$=0.22) and at 0.3% (mean $\kappa$=0.10). As larger particles were less likely to contain rBC or a lower rBC mass fraction in ORACLES (Dobracki et al., 2022; Sedlacek et al., 2022), this may reflect a composition dominated by more hydrophilic species such as sulfuric acid. This variability overall supports existing findings that the accumulation mode is at least partially externally mixed (Dahlkötter et al., 2014; Denjean et al., 2020; Dobracki et al., 2022; Sedlacek et al., 2022; Taylor et al., 2020), which

results in measurable differences in hygroscopicity. This will be supported further by examining hygroscopicity using LASIC data in section 3.2.3. The internal mixing assumption in WRF-CAM5 renders it unable to capture these observed features.

### 3.1.2 Aging processes in the free troposphere, WRF-CAM5 vs. ORACLES

Biomass-burning aerosols emitted in Southern Africa take roughly 4-14 days to be advected to the

remote marine free troposphere, leading to optically thick smoke layers reaching as far west as Ascension Island and beyond (Chand et al., 2009; Zuidema et al., 2016). Over time, particles may undergo drastic physical and chemical changes such as heterogeneous oxidation, fragmentation, coagulation, and photolysis—impacting mass, density, optical properties, or hygroscopicity (Che et al., 2021a; Dang et al., 2022; Dinar et al., 2006; Dobracki et al., 2022). There is consistent observational evidence for a loss of

organics with increasing smoke age and oxidation markers in ORACLES and CLARIFY observations (Che et al., 2022; Dang et al., 2022; Dobracki et al., 2022; Sedlacek et al., 2022). Lab studies have suggested that, on the ~3-14 day timescales relevant to these observations, this loss may be caused by heterogeneous oxidation—especially fragmentation—that functions to re-volatilize and evaporate organics (Che et al., 2021a; Kroll et al., 2009; O'Brien & Kroll, 2019). This configuration of WRF-

CAM5 forms SOA by predefined conversion factors applied to various organic gases such as isoprene



and xylene. The density and hygroscopicity of each separate aerosol chemical species involved is constant.

The aerosol size distribution also evolves through new particle formation, coagulation, and evaporation. Here, we analyze the evidence of some aging processes in ORACLES observations and their
representation, or lack thereof, in WRF-CAM5.

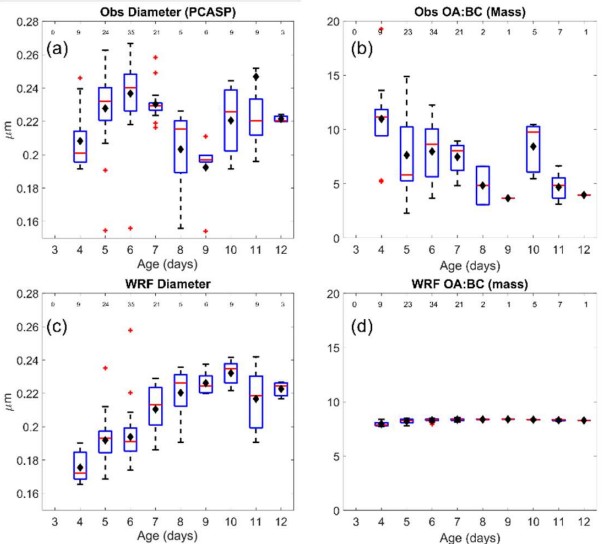

Figure 5: Aging trends in FT for mean diameter (a, c) and OA:BC mass ratio (b, d). Sample sizes for each box-whisker are listed at top of each figure. Observational data are filtered for total aerosol mass>10 μg/m$^3$ and rBC mass>0.1μg/m$^3$, and same subset is then sampled in WRF-CAM5. Black diamonds represent mean, red lines represent median.

Mean particle diameter is a useful indicator of both particle evolution and CCN activity (Kuang et al., 2020). Mean diameter calculated using ORACLES and CLARIFY PCASP instruments shows a non-monotonic change with age, with a general trend towards growth over the 4-6 day range, and then a flattening or decreasing diameter thereafter (Fig. 5a). The PCASP is used here because it was the only available sizing instrument across both ORACLES and CLARIFY campaigns and therefore illuminates longer-term trends than ORACLES alone. The trend of mean diameter growth in the first ~3-7 days is also

captured by the ORACLES LDMA and UHSAS (Fig. A2). However, as ORACLES has very few samples aged beyond ~7 days, the flattening or decreasing diameter trend cannot be corroborated by the more
highly size-resolved instruments here. WRF-CAM5 shows an overall positive trend (Fig. 5c)—the mean diameter grows steadily from approximately 185 nm to 230 nm between 4 and 12 days. This is expected as the model lacks a mechanism to lose OA particle mass over time, while particles can grow through coagulation and secondary aerosol condensation. There is no evidence of wet scavenging in the free troposphere—either in the model or observations—that might otherwise allow new-particle formation to
assert itself in a previously smoky FT air parcel.

Additionally, observations show a noisy downwards trend in the OA:BC mass ratio over time (Fig. 5b), while in the model the ratio is nearly completely flat (Fig. 5d) which implies negligible SOA formation in the model. Further, the mass ratio of OA:CO decreases by 54% between ORACLES and



CLARIFY FT samples, but only decreases by 30% in WRF-CAM5 (not shown). This decrease is to be expected as the smoke dilutes and approaches the background CO concentration in the region, roughly ~60ppb measured during clean periods at Ascension in Aug 2017 (Pennypacker et al., 2020). In contrast, BC:CO decreases very similarly in both observations and the model (14% and 17% decrease respectively). Taken together, OA is likely selectively lost over time in a way that the model does not represent. Quantification of this loss rate and specific causal mechanisms, such as fragmentation or photolysis, have been explored in other field, modeling, and lab studies (Che et al., 2021b; Dobracki et al., 2022; Konovalov et al., 2019; Lou et al., 2020; O'Brien & Kroll, 2019; Sedlacek et al., 2022) and could be implemented and tested in the SEA and compared to these observations to assess improvements and impacts.

## 3.2 Boundary layer performance

The marine boundary layer (MBL) in the SEA presents new observational and modeling challenges that are not present in the free troposphere. The MBL represents a new source of primary and secondary aerosols, in the form of sea spray and dimethyl sulfide (DMS) emissions. Smoke is entrained into the MBL at sporadic spatial and temporal scales and is removed by precipitation in similarly irregular ways that complicate 1:1 comparison (Diamond et al., 2018). The MBL has convective turbulence that leads to stratocumulus formation at the capping inversion, and the MBL close to Ascension Island can transition to being frequently thermodynamically decoupled between the surface layer and cloudy layer (J. Zhang & Zuidema, 2019). All these processes can have strong impacts on the composition and size distribution of aerosols and change how they may interact with clouds.

This section focuses primarily on the LASIC campaign. First, it is worth noting some substantial differences between LASIC observations and the airborne ones used so far (ORACLES and CLARIFY). The LASIC campaign's static nature on Ascension Island means its observations are subject to the whims of meteorology and cannot seek out smoke parcels, as aircraft can. Smoke also only reaches ASI when it has been entrained—either locally or upwind—into the BL.

Second, as Ascension Island is approximately 3,000 km west of Angola, smoke is substantially more aged and diluted in both CLARIFY and LASIC data than the smoke measured during ORACLES. For the purposes of this work, LASIC analysis will be limited to August 2017 since that is when it overlapped with both ORACLES and CLARIFY. It is also worth noting that at 36 km resolution, WRF-CAM5 treated the cells containing ASI as ocean uniformly and so the model includes no meteorological features related to land or topography.



Figures 6a-e show the time series of smoke properties and rain at ground level at ASI. We have identified and labeled periods considered smoky, medium, and clean for the sake of separating smoke properties during this month by regime, based upon tercile concentrations of black carbon similar to Zhang & Zuidema (2019). This section compares WRF-CAM5 modeled properties to observations of the BL aerosol properties, size distribution, hygroscopicity, and mixing state, and concludes with an analysis

of boundary layer dynamics and rain in observations and WRF-CAM5 ASI through the month.



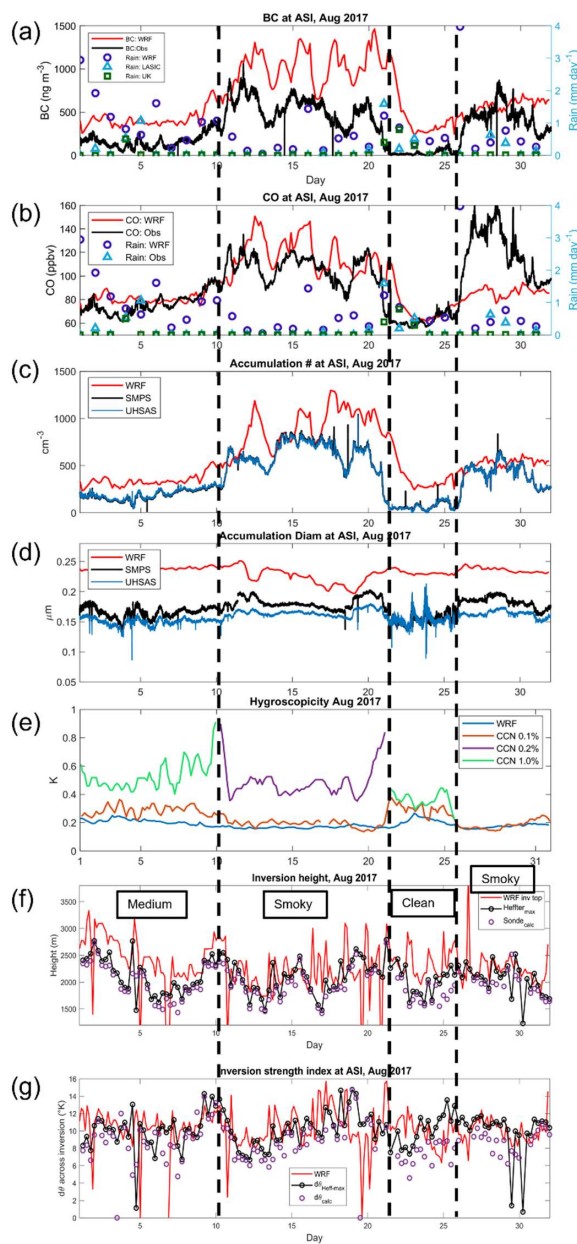

Figure 6: Time series of smoke properties at Ascension Island in August 2017. Vertical dashed lines delineate periods of smoky, medium, and clean conditions. a-b) refractory BC and CO concentration, respectively. Overlaid on both are rainfall accumulation from WRF-CAM5, LASIC, and UK Met devices summed on each day; c) Accumulation mode number concentration; d) Accumulation mode geometric mean diameter; e) Hygroscopicity from the CCN and SMPS from LASIC, and bulk composition in the accumulation mode in WRF-CAM5; f) PBL inversion height from WRF-CAM5, from the LASIC radiosonde VAP, and recalculated from radiosonde matching the algorithm applied to WRF-CAM5; g) inversion strength from WRF, from the LASIC radiosonde VAP, and recalculated from radiosonde profiles using the same algorithm as applied to WRF-CAM5.



### 3.2.1 Boundary layer smoke composition

Observations from both ORACLES and CLARIFY AMS show a large difference in particle composition between the free troposphere and the boundary layer (Fig. 4), a difference which is generally not captured in WRF-CAM5. In particular, WRF-CAM5 does not reproduce the large increase in sulfate fraction in the BL compared to the FT. By mass fraction, sulfate in observations is enhanced from 11% to 26% in the CLARIFY FT to BL, and from 11% to 37% in the ORACLES FT to BL, respectively. Since free tropospheric smoke is chemically similar between observations and the model, this discrepancy in the BL is unlikely to be related to a model misrepresentation of smoke composition itself. It could instead be a combination of higher sulfate aerosol formation in the marine boundary layer—such as from DMS emissions, which are not included in this configuration of WRF-CAM5—and OA removal. There is also observational evidence of regular and frequent occurrence of new particle formation in the upper part of the remote MBL (Zheng et al., 2021) that have been hypothesized to be driven by DMS and thus contain sulfate. These could then subside into the BL and may be a locally dominant source of sulfate and new particles (Clarke et al., 1998). WRF-CAM5 also retains a large dust fraction in the ORACLES-sampling BL that does not appear in observations as described above. This suggests a model bias towards high fine-mode dust generation rates in the natural dust emission scheme, an issue previously identified in these parameterizations (Kok, 2011).

### 3.2.2 Boundary layer size distributions

While WRF-CAM5 shows reasonable representation of FT mean diameters of smoke aerosols, it broadly overestimates the mean diameter of smoke at ASI (WRF: ~200-240 nm; LASIC: 150-190 nm; WRF-CAM5 mean bias of +35% vs. SMPS and +47% vs. UHSAS, Fig. 6d). This is likely due to a lack of particle losses from multiple sources. First, there are potential chemical losses in single particles (see section 3.1.2). Second, there may be a shrinking mean diameter of the aerosol size distribution following aerosol activation into cloud droplets and wet scavenging, in which larger particles are activated and collected more easily. These occur over long distances, as particles in WRF-CAM5 continue to coagulate and grow.

The accumulation-mode number concentrations are overpredicted in WRF-CAM5 by 60% on average (Fig. 6c), excluding the clean period. The bias is the lowest during the smokiest period, with a median bias of 45% and interquartile range of 14-80%. The overestimation bias is far larger during the clean period, over 1,000%. Some of the bias is attributable to the number concentration bias in the FT, as



this smoke with high $N_{AER}$ entrains into the BL (WRF-CAM5 bias above ORACLES and CLARIFY by ~28-38%), and the remainder may be explained by either over-entrainment or removal issues, as discussed below.

The observed number size distribution shows a consistent accumulation mode centered around 180 nm through both smoky and medium periods (Figs. 7a-c) which corresponds to the smoke transferred from the FT (Fig. 3b). During clean periods, observations show a dominant Aitken mode with a mean diameter of 30-50 nm (Fig. 7c), which remains comparable in number to the Aitken mode during medium loading conditions and which is almost nonexistent during smoky periods. As the smoky free troposphere showed nearly no Aitken mode, the BL particles below ~40 nm are likely coming from new particle

formation driven by marine precursors during clean conditions (see beginning of section 3.2.1) (Zheng et al., 2021). We hypothesize that the observed Aitken mode particles observed during clean conditions are gradually lost through either coagulation with the accumulation-mode smoke after it entrains or cloud processing that combines the Aitken and accumulation modes. This could explain why the Aitken mode is present for clean and medium-level smoke but not observed for smoky conditions. In WRF-CAM5, the

Aitken mode tends to be very small in number and broader than observations. This could be due to new particle formation in the model being suppressed by the constant presence of smoke, but also due to





potential inability of models to properly represent new particle formation in pristine marine conditions as found by previous work (Tang et al., 2022).

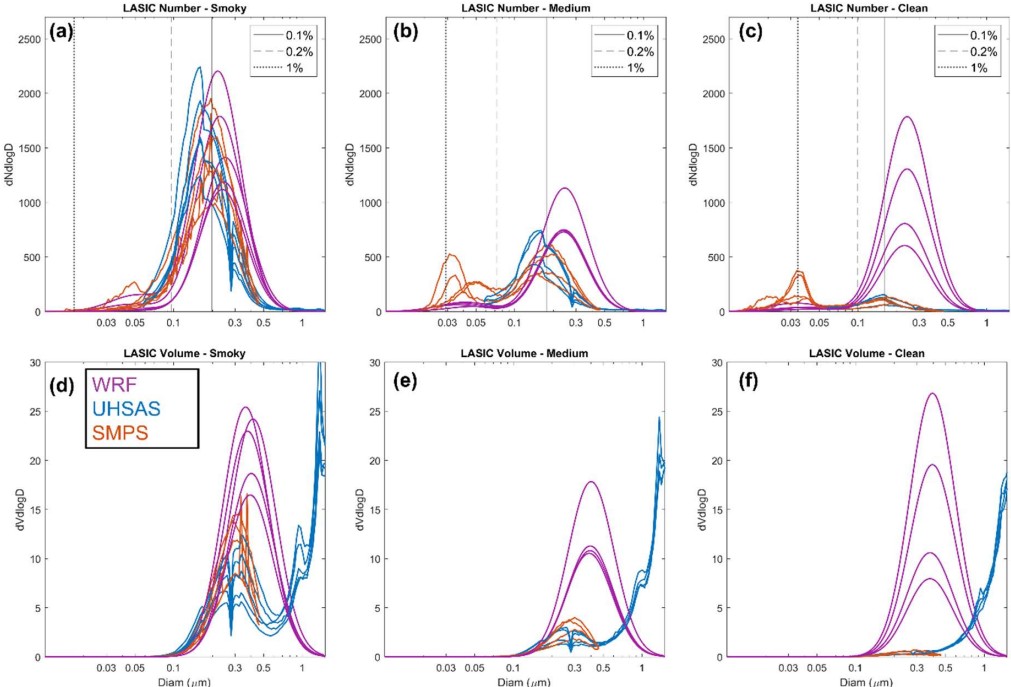

Figure 7: Number and volume distributions from LASIC selected to be representative of the range of conditions during smoky, medium, and clean conditions at Ascension Island. WRF-CAM5 plots show the sum of the accumulation and nucleation mode lognormals.

There is also a persistent population of coarse aerosols through this period as well, predominantly impacting volume. The UHSAS volume distributions at ASI show a large coarse mode above 1 µm regardless of smokiness (Fig. 7d-f). This coarse mode also does not appear in most ORACLES FT data (Fig. 3b-c), suggesting that its emergence at ASI is driven not by smoke. The likely source is sea spray in the MBL (Clarke et al., 1998; Dedrick et al., 2022; Saliba et al., 2019). A caveat in this dataset is that the LASIC ARM emplacement was within ~500 meters of a sea cliff, where winds and breaking waves may

represent a large, localized particle source that is much less influential elsewhere in the SEA BL.

### 3.2.3    Hygroscopicity in LASIC

Estimates of κ based on chemical composition rely on total volume, so the accumulation mode and the coarse mode are the dominant populations impacting chemical κ. However, the number distribution is most relevant to CCN-based κ because it is used to determine $D_{crit}$ at a given SS. Across all conditions,

the $D_{crit}$ at 0.1% SS generally falls in the middle of the accumulation mode, around 170-200 nm (Fig. 7a-




c), and thus we expect that mode to be more representative of bulk smoke κ. $D_{crit}$ at 0.2% SS falls in the range of 75-95 nm, which is in the lower tail of the accumulation mode for smoky periods and tends to be in the overlap region of the nucleation and accumulation mode for clean and medium smoke periods. $D_{crit}$ at 1.0% SS is centered in the Aitken mode (15-35 nm). κ at 0.2% SS has been excluded from Fig. 6e during clean and medium-smoke periods, and 1.0% excluded from Fig. 6e during smoky periods, as the very low number concentration around their respective $D_{crit}$ in these periods leads to highly unreliable κ estimates and eclipses meaningful analysis.

Focusing on the smoky period, LASIC κ at 0.2% CCN supersaturation is larger by a factor of 2 than at 0.1% SS (κ ~ 0.2 at 0.1% SS vs. κ ~ 0.45 at 0.2%). Based on these estimates of κ, the most hygroscopic particles are those near the lower tail of the accumulation mode. Therefore, during smoky periods it may be supposed that these are predominantly sulfate, nitrate, or ammonium particles, or a combination of coagulation and condensation of the same onto the less-hygroscopic BBAs. This is broadly in line with the hygroscopicity of Aitken-mode particles during clean and medium smoke periods, with a similar range of κ. However, it contrasts with FT κ values discussed in section 3.1.1, where κ in the 40-150 nm range is ~0.13, which is lower than in the bulk of the accumulation mode. This suggests processes in the marine boundary layer impact hygroscopicity of the lower tail of the accumulation mode, even in periods of high smoke loading.

WRF-CAM5 closely approximates the CCN-based κ from LASIC at 0.1% supersaturation (SS) and diverges greatly at 0.2% SS. (Fig. 6e). The narrow model variability in κ is explained by the consistent smoky conditions in WRF-CAM5 at ASI through this period, echoing the comparison to ORACLES. WRF-CAM5 also considers particles to be totally internally mixed within each mode, negating the possibility of compositional differences at different size ranges within one mode. With limited chemical evolution and no size-based differentiation possible in each mode, it is reasonable that the model does not produce large hygroscopicity changes. A deeper analysis of observed coating thicknesses and size-resolved particle composition is beyond the scope of this work.

### 3.2.4  Smoke entrainment, removal, and rain at Ascension Island

The period of extremely low BC concentration (< 50 ng m⁻³) observed by the LASIC SP2 between August 20th and 25th is generally not matched by WRF-CAM5. The model shows a median BC concentration bias of +1080% (+280 ng m⁻³) during the same period when shifting by 1 day to account for the time lag vs. observations, and +1950% (+310 ng m⁻³) if matched to observed times directly. However, during medium and smoky periods the BC timing is well-captured, matching the September 2016 findings of Shinozuka et al. (2020). WRF-CAM5 showed a median BC bias of +66% (+330 ng m⁻³) during the



smoky periods and +190% (250 ng m⁻³) during the medium period. This contrasts with the FT, where
WRF-CAM5 does not show a strong bias in smoke BC by either mass (Shinozuka et al., 2020) or mass-
fraction (Fig. 4b). Therefore, the high model bias in BC amount at ASI suggests that the model
overestimates smoke entrainment, underestimates smoke removal in the boundary layer, or both. We
analyze evidence for both possibilities here.

CO is broadly considered a passive smoke tracer on timescales of weeks that is not removed by
wet or dry scavenging of aerosols (Avey et al., 2007; Freitas et al., 2005; Garrett et al., 2010). After a
smoke plume is processed by clouds and the aerosols are largely removed by coalescence and
precipitation, the CO co-emitted with BBAs is expected to remain as a tracer of smoke presence. Thus,
CO is a good tracer to isolate smoke entrainment. Figures 6a-b show a time series of both BC and CO at
Ascension, overlaid with rain measurements. In the clean period, we find that BC remains significantly
higher in WRF-CAM5 than observations through most of August, while for CO the model tracks
observations more closely. This points towards the model likely having unrealistically weak aerosol
removal in the BL. If the main issue was overestimation of smoke entrainment, then CO would show
similar overprediction to BC during the clean period because they entrain together.

Another piece of evidence supporting weak modeled aerosol removal on the BL can be seen by
comparing the first (Aug 10-21) and the second (Aug 26-31) smoky periods (Fig 6a,b). Observed BC and
CO enhancements in these periods are significantly different (e.g., CO in period 2 is larger than in period
1, while BC is slightly less), while the model shows closer BC and CO enhancements for both periods
Subtracting a conservative estimate of 50 ppb background CO concentration, the first and second smoky
period have an observed median BC:ΔCO ratio of 0.0092 and 0.0064 (units: μg/m³ : ppbv) respectively.
The model has a BC:ΔCO of 0.0146 and 0.0160 for the first and second periods, respectively. With no
consideration of background concentration, the first and second periods showed BC:CO ratios of .0085
and .0067 in WRF-CAM5 and .0049 and .0037 in observations respectively. A likely explanation for the
observed behavior is the different degrees of BL aerosol removal in the air masses reaching ASI in these
two periods. A lack of this strong aerosol removal can explain the low degree of BC:CO variability in the
model. These two pieces of evidence, together with the model overprediction of mean diameters in the BL
(section 3.2.2), make a compelling case for concluding that aerosol removal in the BL is likely too weak
compared to reality. Of note, the observed clean period from 21-25 Aug is likely caused by advection of
clean air parcels to the island rather than removal, as evidenced by the unseasonably low CO
concentration (Pennypacker et al., 2020).




To better understand potential wet aerosol removal, we evaluate the model's ability to represent precipitation (Fig. 6a). We find that rain is far more frequent overall in the model than in the two observational datasets. The distribution of 3-hour rain accumulation in the model, on the other hand, skews towards lower rainfall volume in each period than in observations, even when limiting the model rain samples to only include those above the LASIC rain bucket detection threshold of 0.05mm h$^{-1}$. (Fig. 8). This is consistent with the well-known "drizzling problem" of global climate models (D. Chen et al., 2021; Stephens et al., 2010; Trenberth et al., 2003; Trenberth & Zhang, 2018). The underprediction of

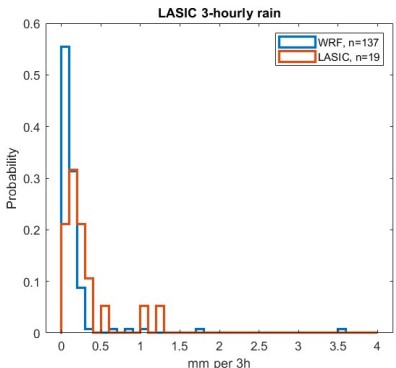

Figure 8: Histogram of 3-hourly rain rate measured by LASIC. In the legend, n represents the total number of rain events sampled over the detection threshold of 0.05mm per hr. Note UK Met rain data is only archived daily and is not included here.

heavy rain events could be one of the reasons explaining weak aerosol removal, although future work is needed to implement parameterizations that may tackle this issue (e.g. Chiu et al., 2021) and evaluate it in the context of aerosol removal.

Entrainment can be modulated by BL height and inversion strength (Karlsson et al., 2010; Wilcox, 2010), and thus are included in this evaluation (Figs. 6f-g). WRF-CAM5 shows reasonably good correlation with LASIC radiosonde observations of these two metrics. The model BL is slightly higher than observations, with a median bias of +220 m (+10%) during this month compared to the Heffter BLH, and +400m (+21%) compared to the

recalculated BLH values based on the model algorithm. When only analyzing the clean and medium smoke loading periods, the bias is higher at +330m or +15% median bias compared to Heffter, and +510m or +27% compared to the recalculation. A deeper BL can result in enhanced smoke entrainment as smoke doesn't have to subside as much to reach the BL top, increasing the availability of smoke to entrain. On the other hand, WRF-CAM5 inversion strength is well represented or slightly overestimated depending on the calculation used, with a median bias of +0.14°K (+1.1%) compared to Heffter and +1.7°K (+21%) compared to the recalculation. A stronger inversion would be expected to lead to less mixing across this boundary and thus less entrainment, opposing potential effects due to a deeper BL (Karlsson et al., 2010; Wilcox, 2010). Thus, given that BL height and inversion strength biases are low and might result in opposite behavior, these don't support a persistent overprediction of entrainment. This is consistent with the timeseries of CO (Fig. 6b), which show a range of behaviors from CO overprediction (e.g., 1st smoky period) to underprediction (e.g., 2nd smoky period), implying a mixed behavior of model entrainment and not necessarily a persistent bias.




### 3.2.5 Aerosol activation and turbulence

ORACLES and CLARIFY took measurements of aerosols and cloud properties at fine scales, in close proximity to both, and with strong controls on sampling location. This avoids some of the assumptions and screening algorithms that add uncertainty to assessments based on remote sensing measurements, as well as providing better vertical resolution and sampling within clouds.

Aerosol activation into cloud droplets is analyzed here by comparing observed and modeled values of both mass-weighted cloud droplet number concentration ($N_C$) and aerosol number concentration ($N_A$) immediately below that cloud, sampled across CLARIFY and ORACLES. A trend visible in WRF-CAM5 that does not appear in either ORACLES (Fig. 9a) or CLARIFY (Fig. 9b) observations is that the modeled clouds have a much higher upper limit of $N_C$. Observations show an upper range of 400-500 cm$^{-3}$ across both campaigns, while WRF-CAM5 attains nearly 1000 cm$^{-3}$. This may be driven by strong updraft turbulence driving high activation as described below.

CLARIFY observations also capture a cloud population with both $N_C < 150$ cm$^{-3}$ and $N_A < 300$ cm$^{-3}$ that was not seen in ORACLES or in WRF-CAM5. This difference between campaigns may be due to the more scattered clouds and more diluted smoke sampled in CLARIFY than in ORACLES. It may also represent a cloud population that is not substantially impacted by smoke, considering the low number concentration. As mentioned in the previous section, WRF-CAM5 has consistently high ($> 400$ cm$^{-3}$) smoke concentrations around ASI throughout August, so it fails to represent the low-smoke cloud interactions observed there.

The ratio of $N_C$ to $N_A$, representing a rough aerosol activation efficiency, is shown in Figs. 9c-d. Median activation efficiency is 0.77 for ORACLES and 0.50 for CLARIFY observations, and 0.66 and 0.64 in the respective WRF-CAM5 samples. The shift in activation efficiency spectra between ORACLES and CLARIFY, as well as aerosol and cloud number concentration spectra, may reflect a change in predominant cloud domain, such as that from stratocumulus to cellular cumulus, that is not well captured in the model (Abel et al., 2020; Diamond et al., 2022; Zuidema, Sedlacek, et al., 2018).



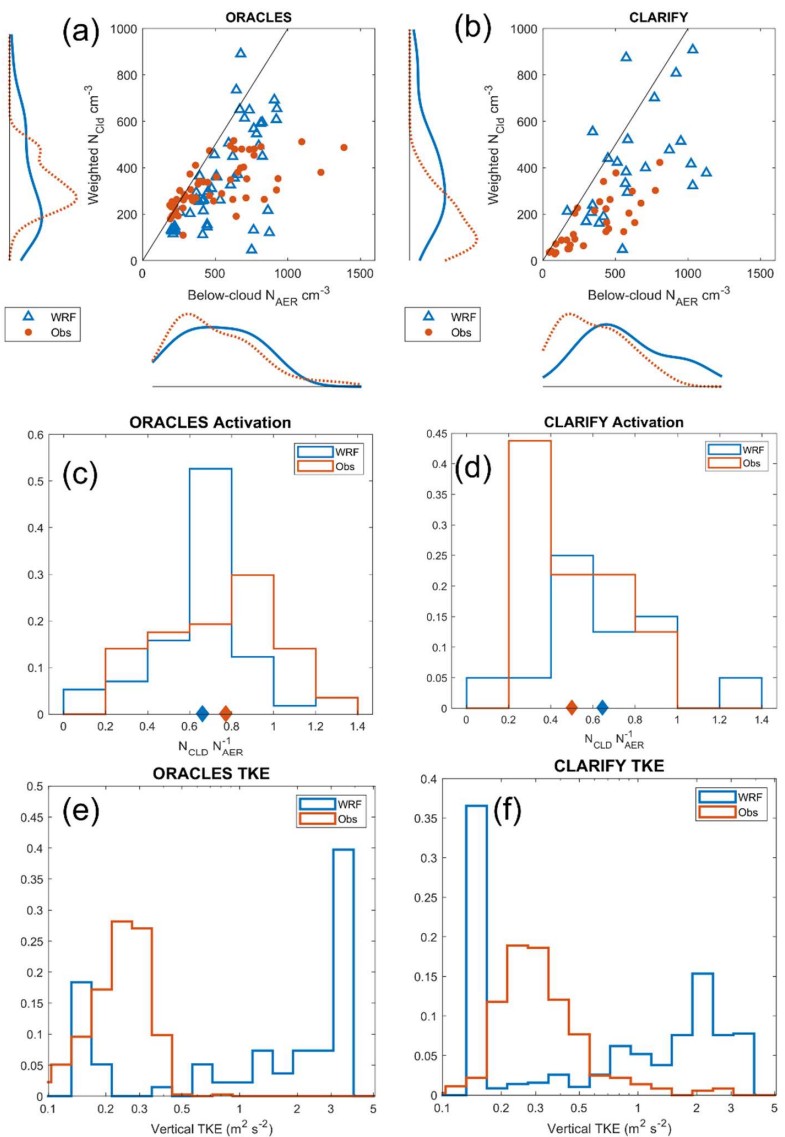

Figure 9: Observed and modeled cloud properties and BL turbulence. a-b) Cloud droplet number (weighted by LWC) compared against below-cloud aerosol concentration from observations and WRF-CAM5 in a) ORACLES and b) CLARIFY cloud transects. Axes of a) and b) show kernel PDFs of each distribution, on the same scale. c-d) Normalized PDFs of activation efficiency, the ratio $N_{CLD}/N_{AER}$ for each campaign + WRF-CAM5. Diamonds on the x-axes represent the median of the like-colored population. e-f) Spectra of BL vertical TKE from each campaign WRF-CAM5 between 100m and 700m. Note: aerosol number concentration in observations is taken from PCASP for consistency across campaigns, which has a lower size limit of ~110nm. This cutoff was virtually imposed on the WRF-CAM5 size distribution as well for this figure.





Turbulent updraft strength is a main driver of the water vapor supersaturation within a lifted parcel,
and thus the activation tendency of an aerosol population (Ditas et al., 2012; Prabhakaran et al., 2020).

Compared to both ORACLES and CLARIFY BL measurements, WRF-CAM5 substantially
overestimates the updraft strength (Fig. 9e-f) and has a bimodal TKE distribution, rather than the
unimodal character of observations. The large peak in TKE distribution near 0.15 $m^2$ $s^{-2}$ in WRF-CAM5
comes from a coded lower-limit on TKE. These strong updrafts could generate a population of
erroneously high $N_C$ if conditions were suitable, which could explain why the model does not capture the

750 observed $N_C$ upper limit. We also note that the spread of $N_C$ in the model is much larger than the
observations ($N_C$ standard deviation in observations=101 $cm^{-3}$; in WRF=219 $cm^{-3}$) while this is not the
case for $N_A$ (observed standard deviation=227 $cm^{-3}$; in WRF=236 $cm^{-3}$), which can also be explained by
overpredicted spread in model TKE. The observed probability distributions of TKE are consistent
between the ORACLES and CLARIFY anemometers despite the large spatial separation and are

755 consistent with values for ORACLES reported by Kacarab et al. (2020).

## 4 Conclusions

This work has analyzed the performance of WRF-CAM5 against the ORACLES, CLARIFY, and
LASIC field campaigns. The goal has been to assess model representation of biomass-burning smoke and
aerosol-cloud interactions in the Southeastern Atlantic Ocean, especially focused on diagnosing process

differences. Previous work, as well as our analyses, show that different instruments on the same platform
and across platforms are often sufficiently consistent to compare jointly with the model, expanding our
analysis and conclusions.

In the free troposphere, WRF-CAM5 captures the average physical and chemical properties of the
younger smoke measured by ORACLES (bias in number, mass, and volume concentrations of +28%, -

765 10%, and -16% to +36%, respectively) with larger biases for older smoke measured by CLARIFY (+38%,
+108%, and 111% for the same metrics), implying issues with model representation of smoke aging.
Mean diameter is captured within variability in the ORACLES observations after increasing the diameter
in the emissions (from 110 nm to 150 nm) to be more consistent with literature values. Although smoke
composition in the FT is well-captured in the model, especially the fractions of OA, sulfate, and BC, we

also find WRF-CAM5 underpredicts hygroscopicity by ~25-35% in the smoky FT (WRF-CAM5 median
0.125-0.136; ORACLES and CLARIFY median 0.172-0.195). This κ bias could be caused by a lack of
$NH_4$ and $NO_3$, overprediction of dust, and misrepresentation of OA properties (e.g., low prescribed
density and kappa, as well as the change in those values changing with age).



Notably, in both ORACLES and LASIC observations, we find that CCN-estimated κ exhibits a large
range for smoky conditions across different particle sizes in the 20 nm-300 nm range. FT (ORACLES)
smoke shows a lower κ in the lower tail of the accumulation mode compared to the center (κ ~0.1 vs. ~0.3
respectively) likely due a larger fraction of black carbon at lower sizes.

By comparing mean smoke properties using modeled age estimates in the FT, we find that WRF-
CAM5 is likely missing significant aging processes impacting smoke mean diameter and composition.
The OA:BC mass ratio as well as the OA:CO and BC:CO ratios compared across 4-12 days of transport
show that OA is selectively being removed and therefore limiting particle growth, which is not
represented by the model. This process is a valuable target for future work since current literature
studying smoke aging beyond several hours is limited, and because simulated particle size can impact
aerosol-cloud interactions and estimates of cloud radiative effects in the region.

Observations of aerosol composition in the boundary layer show a large relative enhancement of $SO_4$
in the compared to the FT, increasing from 11% (ORACLES FT) to 37% (ORACLES BL) and from 11%
(CLARIFY FT) to 26% (CLARIFY BL), neither of which are represented by the model. This suggests
WRF-CAM5 has missing or weak processes that lead to sulfate aerosol in the MBL, such as BL ocean
DMS emissions (not included in this model build) and smoke removal, both of which allow for clean
periods of sulfate particle formation. During clean and medium-smoke loading periods, the LASIC SMPS
also shows an Aitken mode that is likely driven by new particle formation and has hygroscopicity values
similar to sulfate.

Hygroscopicity in MBL (LASIC) smoke, similar but opposite the trend in the FT, varies between the
lower tail of the accumulation mode and its center (κ ~0.5 vs. ~0.2). This is likely caused by sulfate
uptake of smaller particles through coagulation of the Aitken mode or precursor condensation. This
suggests significantly different chemical composition at different sizes and thus some external mixing
within the accumulation mode. That this trend is apparent at very different smoke ages and locations
suggests that it is a consistent feature of smoke aerosols, and one which WRF-CAM5 is not able to
simulate due to its modes being internally mixed. This should be considered as a mitigating factor in
future studies of BBA hygroscopicity and composition, as both are highly size-dependent.

Overprediction of aerosol concentrations by the model are further enhanced in the MBL when
comparing to LASIC observations, with biases that increase as conditions go from smoky (model bias
+14-80%) to clean (model bias +1000%). While this behavior could be explained by either too-strong
smoke entrainment or too-weak aerosol removal in the MBL, multiple pieces of evidence point to the
latter being the primary factor. First, mean aerosol diameter substantially decreases in observations from



180-240nm FT in the CLARIFY and ORACLES FT to 140-180 nm in the LASIC MBL. The model
shows little change in mean diameter. This points to cloud processing of aerosol rather than a smoke
process on its own. Comparing the behavior of BL CO and BC concentrations can further provide
insights. Also, observed BC to $\Delta$CO ratios, assuming a background of 50 ppb CO, change substantially

between the two heavily smoky periods (BC/$\Delta$CO=0.0092 in the first, and 0.0064 in the second), which
can be explained by differences in BC removal across the history of these airmasses. This variation is
weaker in the model (BC/$\Delta$CO =.0146 in the first smoky period, 0.0160 in the second). We also find that
WRF-CAM5 has rain that is far more frequent, though lighter, than observations support—in line with the
known "drizzle problem" of GCMs—which could contribute to a weak aerosol removal. Finally, model

evaluation of inversion height, inversion strength, and MBL CO show modest biases (+10-21%, +1.1%,
+0.5% mean biases respectively) that also oppose each other in illustrating entrainment tendency, and
overall do not support a persistent overestimation of entrainment. These clean conditions are often when
aerosols may have the largest relative impact on cloud number (Kacarab et al., 2020) and are especially
important to constraining aerosol-cloud radiative forcing (Gryspeerdt et al., 2023). An inaccurate

representation of aerosol removal and smoke-free conditions should therefore be taken into account for
future modeling analyses of aerosol-cloud-radiation interactions.

The activation ratio for below-cloud aerosols (0.1-3 µm) into liquid droplets is relatively constant
in WRF-CAM5 samples in both ORACLES and CLARIFY at $N_{CLD}/N_{AER}$ ~0.65. However, observations
show a higher activation tendency in ORACLES ($N_{CLD}/N_{AER}$ ~0.78) and lower in CLARIFY ($N_{CLD}/N_{AER}$

~0.5). Observed $N_C$ in both aircraft campaigns shows an upper limit of ~400-500 cm$^{-3}$, which is exceeded
occasionally by WRF-CAM5 by 300-500 cm$^{-3}$ across both campaigns and leads to a wider modeled
spectrum of $N_{CLD}$. Vertical TKE was analyzed using both ORACLES and CLARIFY anemometers.
WRF-CAM5 is found to overestimate TKE by up to a factor of 10 in the boundary layer compared to both
campaigns, as well as showing a bimodal distribution rather than the observed unimodal distribution. The

strong model turbulence may contribute to the model exceeding the upper limit of observed $N_{CLD}$ and
overpredicting the $N_{CLD}$ spread.

The performance of WRF-CAM5, despite its biases and missing processes, represents a useful
tool for the study of smoke aerosols. LASIC, CLARIFY, and ORACLES present an especially rich suite
of observations against which to measure model representations of major atmospheric processes such as

boundary layer turbulence, smoke composition and size changes over long aging periods, and aerosol-
cloud interactions. Previous analysis has shown that WRF-CAM5 performs well in representing regional
smoke advection and mixing trends, as well as smoke plume positioning and bulk optical properties at
least in the lower FT (Shinozuka et al., 2020). Schemes allowing OA removal over aging timescales of



~14 days may substantially improve composition and bulk optical properties in models and thus need to be tested in future work. Sulfate representation in the MBL may also be improved both by improving DMS emission schema, and through improvements in scavenging schemes that allow for ultra-clean regions to emerge and lead to significant new particle formation.

The impact of smoke evolution on cloud droplet nucleation is highly variable and remains difficult to model in GCMs, which commonly have simple aerosol evolution schemes and aerosol mixing state assumptions, as well as frequently coarse resolutions of ~0.25-1 degree. If TKE spectra may be improved, then a more accurate aerosol chemistry and mixing state schema and better representation of aerosol removal in the MBL may improve cloud microphysical properties, which could help reduce uncertainties in modeled aerosol-cloud radiation interactions. Future work could use similar methodology presented in this work to evaluate other modeling systems to assess if similar biases are present and

implement model improvements. Finally, an assessment on how these improvements modify effective radiative forcings and climate impacts of smoke should be performed.

*Appendix A*

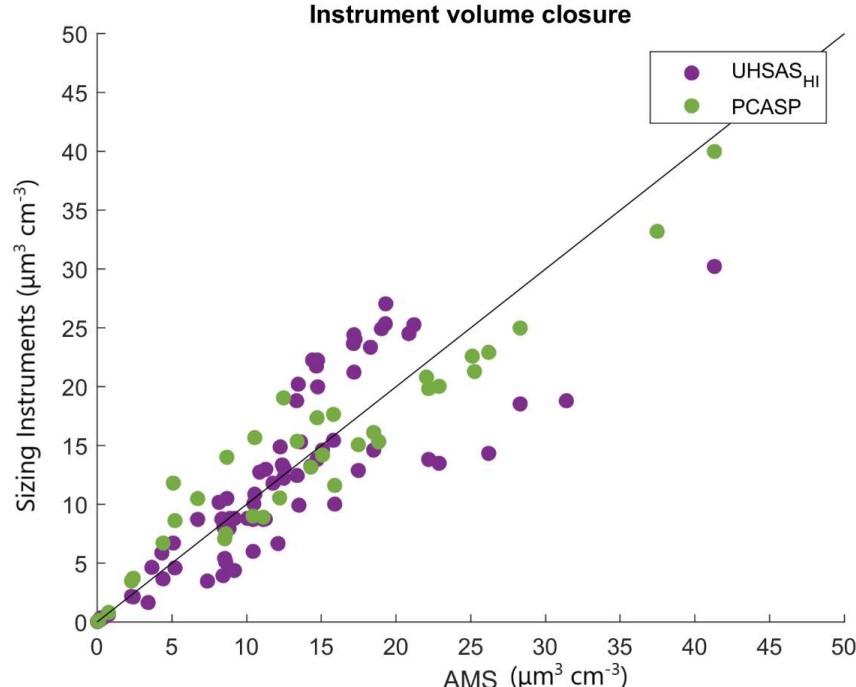

**Figure A1**: Total volume concentration in the ORACLES free troposphere, comparing both the U. HI
UHSAS and PCASP each against the AMS. Densities assumed for the AMS are listed in Table 2 of main text.





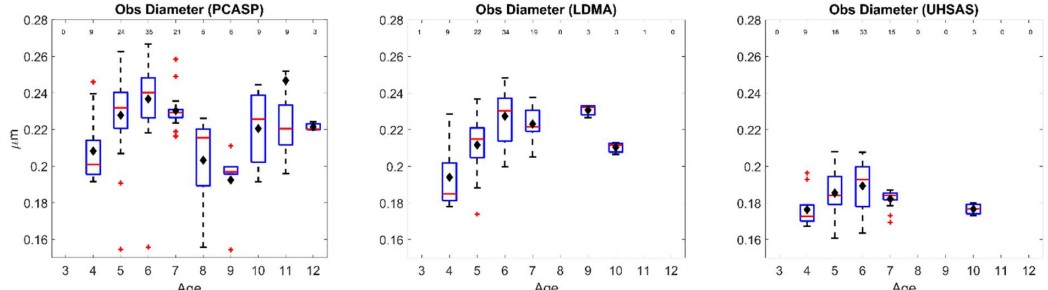

**Figure A2**: Geometric mean diameter from observations, binned by WRF-AAM average plume age. PCASP plot uses samples from both ORACLES and CLARIFY, as the only aerosol sizing instrument available in both campaigns. LDMA and UHSAS are both only from ORACLES samples.

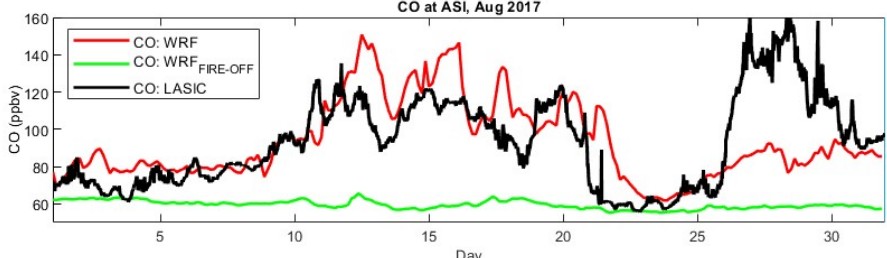

**Figure A3**: CO concentrations from WRF-CAM5 both with and without QFED2 fire emissions to illustrate model background, and observations from LASIC for August 2017.

*Dataset Availability*

VIIRS map available at: NRT VIIRS 375 m Active Fire product VNP14IMGT distributed from NASA FIRMS. Available on-line https://earthdata.nasa.gov/firms.

doi:10.5067/FIRMS/VIIRS/VNP14IMGT_NRT.002

Observational datasets for ORACLES 2017, as well as colocated WRF-AAM plume age estimates, are available through the NASA ESPO data archive: https://espo.nasa.gov/ORACLES/archive/browse/oracles/id14/P3.

Observational datasets for CLARIFY-2017 are available at through the CEDA data archive: https://catalogue.ceda.ac.uk/uuid/38ab7089781a4560b067dd6c20af3769

Datasets for LASIC are available individually as follows on the ARM data archive. Data accessed between 01 Aug 2018 and 02 Feb 2022:

Weighing Bucket Precipitation Gauge: http://dx.doi.org/10.5439/1338194



Ultra-High Sensitivity Aerosol Spectrometer: http://dx.doi.org/10.5439/1333828

Scanning mobility particle sizer: http://dx.doi.org/10.5439/1225453

Cloud Condensation Nuclei Particle Counter (Column A): http://dx.doi.org/10.5439/1323892

Cloud Condensation Nuclei Particle Counter (Column B): http://dx.doi.org/10.5439/1323893

Condensation Particle Counter (CPCF): http://dx.doi.org/10.5439/1352536

Radiosonde Planetary Boundary Layer Height: http://dx.doi.org/10.5439/1150253

*Author contributions*

CH and PS designed the model–observation comparison PS acquired the resources to support this
research. AN, AND, SF, GM, HC, JMH, SGH, CK, SG, MK, AN, JR, AJS, KLT, RW, JZ, JU, and PZ
provided data from instruments during the ORACLES, LASIC, and CLARIFY observation periods. CK,
HW, JMH, JZ, PZ, SF, SG, SGH, and JU assisted with further analysis of observational data. CH led the
model and observational data processing for this comparison with scripting assistance from PS. PS and
CH ran the model and implemented configuration changes. CK, JPSW, JU, LRL, and YZ provided
substantial components of model configuration. CH wrote the first draft. PS provided major input
throughout writing and AJS, AND, GM, JR, JZ, LRL, PZ, RW, SF, SG, SGH, YZ provided further
editing and feedback.

*Competing interests*

PZ and JH are guest editors for the ACP Special Issue: "ACP special issue: New observations and related
modelling studies of the aerosol–cloud–climate system in the Southeast Atlantic and southern Africa
regions." The remaining authors declare that they have no conflict of interest.

*Acknowledgements*

ORACLES is a NASA Earth Venture Suborbital-2 investigation, funded by the US National Aeronautics
and Space Administration (NASA)'s Earth Sciences Division and managed through the Earth System
Science Pathfinder Program Office.

*Financial Support*

Financial support for this work was provided by NASA ORACLES grant 80NSSC19K1463 to PS and
DOE LASIC grant DE-SC0018272 to PZ and PS. JZ was supported by DOE LASIC grant DE-
SC0021250. SG was supported under the NASA Earth and Space Science Fellowship (grant nos.
NNX15AF93G and NNX16A018H). GM and SG were supported by NASA (grant no.



80NSSC18K0222). YZ is supported by the U.S. NOAA Office of Climate AC4 Program
(NA20OAR4310293).

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
