# Peer review of "Biomass-burning smoke properties and its interactions with marine stratocumulus clouds in WRF-CAM5 and southeastern Atlantic field campaigns"

_EGUsphere, 2023_

## Referee Comment (RC3)

Review of

*"Biomass-burning smoke properties and its interactions with marine stratocumulus clouds in WRF-CAM5 and southeastern Atlantic field campaigns"*

by Howes et al. (2023)

**General comments**

This study compares aerosol and cloud properties observed during three separate field campaigns and simulated by WRF-CAM5 over the southeastern Atlantic during the biomass-burning season. Due to the wide coverage of stratocumulus clouds over the region and therefore the strong radiative forcing, understanding aerosol and cloud properties and their interactions over the area is crucial for the climate model improvements. The study has revealed many key discrepancies between observations and WRF-CAM5; for example, the lower hygroscopicity $\kappa$ in the simulations, the lack of aerosol loss processes in the model that may have led to an erroneous aging process and/or number concentrations, and the rainfall tendencies (i.e., the drizzle problem). These points are important for the future aerosol/cloud model developments, and hence the study conveys important messages. Figures and tables present simulated and/or observed results well, which get explained thoroughly in the main text. The structure of the paper and the presentation of the results can be somewhat improved before the publication, as it may be confusing in the current format. While this is understandable given the number of different variables presented from many field campaigns and instruments, I suggest the re-organization of the paper structure so that a few most important messages stand out, rather than keeping them as one of many findings, which I believe will convey the messages more easily to the readers.

**Specific comments**

**General structure**: Given the amount of information in the paper, I believe the clear structure of the paper is crucial for readers to understand the content as much as possible. Currently the main section is structured as follows;

3. Results

    3.1 Free troposphere smoke

        3.1.1 Physical properties

        3.1.2 Aging processes in the free troposphere, WRF-CAM5 vs. ORACLES

    3.2 Boundary layer performance

        3.2.1 Boundary layer smoke composition

        3.2.2 Boundary layer size distributions

3.2.3 Hygroscopicity in LASIC

3.2.4 Smoke entrainment, removal, and rain at Ascension Island

3.2.5 Aerosol activation and turbulence

For example, before starting 3.1 (right after "3. Results"), the authors can explain that the results are presented firstly for the free troposphere and secondly for the boundary layer, and why the two are presented separately. This will give readers an idea of the overall structure and the reasons for the separation. The same thing can be done right after "3.1" and "3.2", so that readers understand why the subsection is split into multiple subsubsections, and what contents are expected within the subsection. This also applies to Section 2 (Methods) as well; the authors can add a few sentences between "2. Methods" and "2.1 Observation systems" on what each subsection contains and why the section is split into five subsections.

The subsection title for 3.1 and 3.2 should be comparable, for instance, "Free troposphere" and "(Marine) Boundary layer". Once these are defined in the subsection titles, then the subsubsection titles do not need to include "free troposphere" or "boundary layer" again (as it's obvious). Ideally, 3.1.1 and 3.2.1 have comparable titles and contents, for example "Physical properties" for both 3.1.1 and 3.2.1, so that each subsection has the same/similar subsubsections and similarly structured. While this may be some work and re-organization of the contents, I believe the paper will convey the messages better to readers. It would be even better if each subsubsection has a clear message or finding, which gets summarized at the end in Conclusions (and also in Abstract).

**Abstract**: Related to my general comments above, the abstract seems to include too much detailed information on many topics, but it should be a concise summary of a few key findings of the paper. In addition, in the current format, readers may get the impression that aerosol size and composition in the free troposphere are the only variables that the model was able to simulate well, and the rest was not well simulated/reproduced, though this is not the main point of the paper. I believe summarizing a few key discrepancies and their cause in brief words will give a better idea of the paper content.

**Figure 2**: The red crosses in the plots are not explained, but I believe these are outliers? Please add this information to the figure caption.

**Line 418**: Should this be "above" 3 μm rather than "below", if the topic is the lack of coarse-mode particles?

**Line 588**: "excluding the clean period" – why was it excluded? From the figure, the overprediction is also seen during the clean period.

**Conclusions**: I suggest that the authors list the key findings either in bullet points or in numbers, and make them concise. The detailed numbers do not need to be repeated here for all the items, except for a few key important numbers.

**Figures 2g-h, 4b-I, 6f-g**: Some of these figures are not referred to in the manuscript, or their contents are discussed but they are not exactly referenced/referred to in the manuscript.

**Figure A3**: Is this figure referred to in the manuscript?

**Technical corrections**

**Abbreviations**: Throughout the paper, there are a few words that are defined earlier (e.g., ASI, FT, SEA, MBL) but the full words are used again afterwards, or defined twice. I suggest that the authors make sure abbreviations are defined only once, and they're used throughout the paper after the definitions.

**Line 73**: "Intergovernmental", not "International".

**Line 238**: Add "(2022)" after "Barrett et al."

**Line 268**: Missing the reference for the WRF-Chem model.

**Line 272**: Delete "Niemand et al., " in the bracket, as only the year is necessary here.

**Line 427**: I believe this is a typo, the "μ" at the end needs to be deleted.

**Line 516**: This should be "downward" without an "s".

**Line 563**: I believe it should be either "…, which is generally" or "…, (which is) a difference that is generally"

**Line 595**: It should be either ", which corresponds to…" (with a comma) or "that corresponds to…"

**Line 598**: Remove "which"

**Line 598**: "smoke-free" instead of "smoky free"

**Line 635**: Add "к" between "than" and "in"

**Line 671**: Add a period after "periods" to end the sentence.

**Line 673**: "periods" instead of "period"

**Line 721**: I think it should be "provides" instead of "providing", assuming that the subject is "This" in the beginning. Or, it should be "…, while providing…".

**Figure 9 caption**: Add "and" between "each campaign" and "WRF-CAM5" (6th line).

**Line 777**: Add "to" after "due".

**Line 786**: Add "MBL" before "compared to".

**Line 797**: The beginning of this sentence needs to be corrected – I believe it is "The fact that"?

**Line 817**: "are often when" should be "often correspond to the time period when" or something similar.

**Line 834**: Replace "measure" with either "compare" or "evaluate".

**Line 836**: This sentence is more appropriate in Introduction, not in Conclusions.

**Line 886**: Add a dot after "comparison".

---

## Author Comment (AC1)

The authors thank the referees for their time and thoughtful feedback, which have significantly improved the text and substance of the final manuscript. Replies from authors are given below in blue.

**Reviewer 1:**

This manuscript evaluated the performance of WRF-CAM5 against three field campaigns: ORACLES, CLARIFY and LASIC, focusing on model representation of biomass-burning smoke over the Southeastern Atlantic Ocean. Many interesting points are made regarding to the model biases such as likely missing or too slow smoke aging processes, missing or weak sulfate aerosol processes in the MBL, lack of chemical composition variation at different sizes due to the internal mixing assumption, too weak aerosol removal due to wet deposition, overestimation of TKE and different spectrum distribution compared to the observations. These results provide valuable insights to the future improvement of biomass burning smoke aerosols in GCMs, and demonstrate a nice connection between GCMs and field campaign measurements.

The analyses were well-done and the manuscript was generally well written. I would recommend accept this paper with some revisions. My specific comments are listed below:

Introduction: Maybe add one paragraph to briefly introduce the three field campaigns. It reads like missing something with only one of the campaign (ORACLES) is mentioned in the introduction. This can be very brief and you can keep more details in Section 2.1.

- We've added a brief description of each campaign in the introduction:
- "Valuable observational constraints on these processes come from three field campaigns in this region overlapping in August 2017. ORACLES (ObseRvations of Aerosols above CLouds and their intEractionS) was a NASA aircraft campaign in 2016-2018 that studied biomass-burning smoke and clouds in the southeast Atlantic using remote sensing and in situ instruments (Redemann et al., 2021). CLARIFY-2017 (CLoud–Aerosol–Radiation Interaction and Forcing: Year 2017, Haywood et al., 2021), was a UK Met Office aircraft campaign based primarily around Ascension Island in the southeastern Atlantic, and was also studying physical, chemical, and radiative effects of biomass-burning smoke in this remote region. Finally, LASIC (Layered Atlantic Smoke Interactions with Clouds, Zuidema et al., 2018) was a US Department of Energy campaign that installed Atmospheric Radiation Measurement (ARM) Mobile Facility 1 on Ascension Island to observe the remote marine troposphere in both 2016-2017, covering both years' biomass-burning seasons."

Line 141: WRF-Chem or WRF-CAM5? this is sometimes confusing readers that may think there are two model versions are used in this paper. Maybe one can use "WRF-CAM5 with full chemistry" to replace WRF-Chem or declare that "WRF-CAM5 is build on WRF-Chem with

CAM5 physics packages" when WRF-CAM5 is firstly introduced. I don't know which way is more official.

- This was a mistake and has been corrected to say WRF-CAM5.

Line 208: UHSAS usually has a lower size limit of 60nm, and the measurements in the lowest few bins have larger uncertainty, so the effective Dcrit is even larger. This may cause unreasonable k values, especially in large SS or high-k conditions. I saw the authors had some discussions later, it may be good to move some discussions up front here (e.g., what is the reasonable range of calculated k? CCN in SS=1.0% are not used due to too low Dcrit)

- We've added a comment at this line that the ORACLES CCN only measured up to 0.3% SS. Later in the paper Fig. 3 shows that the ORACLES critical diameter is still well above 60 nm at 0.3% SS, and remains well within the center of the accumulation mode. The large uncertainty in κ at small Dcrit for LASIC is also noted in section 3.2.2. 1.0% supersaturation is only analyzed from LASIC data when there is a prominent Aitken mode (i.e., not during smoky and medium conditions), as otherwise the large uncertainty makes κ calculations unreliable. Additionally, the SMPS is used for LASIC which samples down to 10 nm.

Moreover, the inconsistency of aerosol size distribution and CCN concentration measurements may also cause unrealistic k value (when total aerosol number from size distribution is greater than CCN concentration). So, how many valid k data are obtained? Did you systematically exclude high hygroscopicity case?

- Aug 15$^{th}$ UHSAS data was not used for exactly this reason, but was the only day where number counts vs. CCN count at 0.3% SS were physically unreasonable. We've added this text to this section to explain:
- "UHSAS and CCN data is not used for 15 August 2017, as it was found that CCN counts at 0.3% SS for that day exceeded the UHSAS count, which is not physically realistic. Number concentrations and D$_{crit}$ on the other days are within plausible ranges of count and derived κ. Kacarab et al. (2020) similarly found CCN Dcrit in the 100-200 nm range in ORACLES data, supporting this assessment."

Line 315: what is the aerosol initial condition? How long did you spin up your model? Aerosols are known to have days or even longer spin-up time even without considering cross-hemisphere transport in global model (then the spin up time will be months). Just wondering if the initial spin up time is enough to allow African emissions can be transported to the targeted region. I like the idea of reinitializing meteorology every five days but carry over aerosol conditions.

- We've added these comments on the regional spinup period and initial conditions to clarify:
- "The model run period starts July 15, 2017, and is run through August 31, 2017. The July portion is discarded as meteorology and emissions spin-up time, but it allows smoke to circulate through the SEA region. Initial and boundary aerosol and chemical concentrations come from CAMS."

Line 394-402: it may be better to put these numbers in the figure.

- We've rebuilt fig 4 and corrected references to it, so that hygroscopicity is shown for both the FT and BL, both ORACLES and CLARIFY, and in WRF both with and without both Chl + Dust, in order to match AMS as closely as possible (addressing comment below).

Line 424-425: this sentence "As noted above these likely are a minimal mass component in observations" is pretty vague and I don't understand what it means.

- We've clarified how this relates to the closure calculations between the AMS and PCASP, namely that the particles observed by the AMS (which here do not include refractive mineral dust nor chloride) accurately capture the great majority of the aerosol population in these cases.

Line 435: Its unfair that you exclude sea salt, which has the highest k value, but keep dust, which has a very low k value, from WRF-CAM5. Both seasalt and dust are not detected by AMS, why do you only exclude seasalt?

- See response above to comment for line 394-402.

Line 436: I don't understand this sentence, may need to rephrase it.

- We've rephrased the sentence from:
- "The increase in AMS-based $\kappa$ in the BL tracks with the elevated sulfate in the BL compared to the FT across both campaigns."
- To instead read:
- "The higher sulfate in the BL compared to the FT drives the corresponding higher BL $\kappa$."

Line 569-570: it may be more logical to state it in an opposite way: "… a combination of lack of sulfate aerosol formation and weaker OA removal in WRF-CAM5", as observation is treated as the truth and this represents model biases.

- We've rewritten the sentence in question to read:
- "It could instead be a combination of WRF-CAM5 having weaker sulfate aerosol formation in the MBL—with this WRF-Chem build not including DMS emissions—as well as a lack OA removal."

Line 576: why it does not suggest low deposition rate here?

- In the model, fine dust is in the accumulation aerosol so settles the same way as other aerosol in this mode. This sentence was modified for clarity to:
- "This suggests a model bias towards high fine-mode dust generation rates in the natural dust emission scheme, which is an issue previously identified in dust parameterizations (Kok, 2011)."

Line 682: unseasonably à unreasonably? and why is it "unreasonably" as the authors already provided a reasonable guess of the cause.

- We've clarified this comment to say that the clean-period value is very low for this time of year. This clean period is likely to be caused by a pristine airmass, rather than one which mixed with smoke that was later scavenged out.

Line 723: is Na also mass weighted? if not, please add "mean" or "median" before it; if yes, please explain why it is needed for aerosols.

- We've added 'average' to differentiate $N_A$, which is not weighted.

Line 724: what "trend" in WRF-CAM5? Much higher upper limit of Nc is a "model issue" or "bias".

- That phrasing has been corrected to 'bias.'

**Reviewer 2**

This paper presents an evaluation of WRF-CAM5 against observations from three southeastern Atlantic field campaigns. Good results are obtained: biomass burning aerosol size and composition are generally well captured in the free troposphere. Aerosol diameter in the model increases as smoke ages, contrary to observations, likely due to lack of photolysis and heterogeneous chemistry. The observed enhancement in sulfate in MBL relative to FT is underpredicted, and new particle formation is not strong enough, yet aerosol concentrations in the MBL are still overpredicted, as is CDNC. Activation fraction is also overpredicted. There is an interesting analysis of hygroscopicity and BC:CO ratio.

The paper is well written and, in my opinion, at the forefront of current research in this area. I recommend it be accepted after the following minor comments are addressed.

Minor comments:

Abstract line 41: it could be made clearer that the 'broad swath of aerosol properties' does not include any optical properties relevant to aerosol-radiation interactions. These have been evaluated in WRF-CAM5 extensively in other studies, so it makes sense not to include them, but still it may be worth better delineating the scope of the paper.

- We've edited this line to read: "…evaluate a large range of the model's aerosol chemical properties, size distributions, processes, and transport,…"

Introduction is good, and sets the paper in the context of previous work well. However, it could do with a little more perspective on the importance of good representation of aerosol-cloud

interactions in the SEA, perhaps emphasizing the potential role of aerosols in modulating the stratocumulus-to-cumulus transition.

- We've added this to the introduction regarding the value of good model representation in this region:
- "Aerosol-cloud interactions in the SEA can drive large regional uncertainty in radiative effects through multiple mechanisms. Absorbing aerosols in this region have been, to varying degrees, connected to changes in cloud albedo, fraction, lifetime, drizzle rate, cloud droplet size and number, and large-scale breakup or persistence (Christensen et al., 2020; Diamond et al., 2022; Yamaguchi et al., 2015, 2017; Zhou et al., 2017). Therefore, constraint on both smoke representation in models, and especially aerosol-cloud interactions, is crucial to reducing uncertainties in global climate projections. "

L208: since the $(24/D_{crit})^3/(SS\%)^2$ equation does not appear explicitly in Petters & Kreidenweis 2007, it would be helpful to explain briefly where it comes from, and what are the units of $D_{crit}$?

- We've added the text below to explain. The written equation is a rewriting of Eq. 10 in Petters and Kreidenweis with a reasonable linearization of the logarithm (which has an error of less than .1% for supersaturations between 0.1% and 2%), as well as neatly matches the lookup table on Markus Petters's website (https://mdpetters.github.io/hygroscopicity/) :
- "This equation is based on eq. 10 in Petters & Kreidenweis (2007), takes $D_{crit}$ in nm, substitutes numerical values for the constants suggested, and approximates $\ln(1+SS) \sim SS$, for realistic supersaturation values of 0.1%-1.0%."

Is there a reference for using 0.79*sigma_w as vertical TKE?

- We've added a sentence to clarify that the 0.79 factor is part of the Morales & Nenes derivation as well.

Were the UK/CLARIFY rain gauge and the LASIC rain gauge at different locations on Ascension Island? If so where/how far apart? Or were they co-located? Should we expect them to agree?

- Added this to the end of the Data Processing section:
- "Two rain gauges were used for LASIC to help account for orographic lifting potentially impacting rain rates at the ARM station (Zuidema et al., 2018). The ARM station was situation in the more mountainous and elevated eastern half of the island (7.967S, 14.350W). The UK Met Office rain gauge was located at the UK air base and meteorology station approximately 6km to the west, in a relatively flat region of the island (7.967S, 14.4W). Thus, the differences between them are to be expected and are not driven by instrument uncertainty."

Model description: since CDNC are a topic of the paper, it would be helpful to add how the model handles cloud microphysics (I assume Morrison & Gettelman 2008 from CAM5). May be worth mentioning that the use of the Fountoukis & Nenes activation parameterization as

referenced in Zhang et al 2015 is a departure from standard CAM5? What about aqueous-phase chemistry, which seems likely important to explain the increased sulfate in the MBL in observations discussed later?

- We've updated section 2.4 to note that CBMZ is also undertaking aqueous aerosol chemistry, noted that the FN activation is not in standard CAM5, and that we're using CAM5 standard cloud microphysics (Morrison & Gettelman 2008).

How is the TKE used in the activation parameterization? I couldn't immediately find this out from Zhang et al (2015). Presumably its square root is added to the grid-scale updraft, with some scaling factor?

- We have clarified in the description of TKE derivation that it is used in combination with grid-scale updrafts to construct an updraft spectrum, and activation is calculated integrally over that spectrum.

Figure 3b,c are hard to read, axis labels are too small and axis ranges are too large. I can't make out any differences between any of the instruments or the model, I suggest splitting the figure into a few subfigures to spread out the data.

- We've made the axis labels and numbers a larger font, as well as redesigned the figures. The mean of each data source is now a darker, thick line, with the spread of underlying distributions as thinner, lighter lines behind it. This illustrates as concisely as possible what the mean behavior looks like, while keeping some of the variability visible.

Line 427 extraneous "mu".

- Removed.

Line 440, Section 2.2, Table 2: more accurate hygroscopicities for some components, for example sulfate and ammonium nitrate, could be obtained from Figure 2 of Schmale et al (2018), https://acp.copernicus.org/articles/18/2853/2018/ and references therein. Suggest update analysis (if possible) – it seems the 'AMS kappa' values will be somewhat underestimated.

- The analysis in this manuscript uses assumed hygroscopicity values that are within a common range compared to recent model/observational studies of biomass-burning smoke [Bougiatioti et al., 2016; Doherty et al., 2022; Eghdami et al., 2023; Kacarab et al., 2020; Wu et al., 2020]. We have updated the conclusions to add the possibility of future work running a sensitivity test to new Kappa values in the model, as well as assuming higher per-species hygroscopicities:
- "A future sensitivity study using newer $\kappa$ values for the AMS and the model—such as from Schmale et al. (2020), generally significantly higher than those used here—could provide further insight into the importance of $\kappa$ and chemical composition in cloud activation."

**Reviewer 3**

**General comments**

This study compares aerosol and cloud properties observed during three separate field campaigns and simulated by WRF-CAM5 over the southeastern Atlantic during the biomass-burning season. Due to the wide coverage of stratocumulus clouds over the region and therefore the strong radiative forcing, understanding aerosol and cloud properties and their interactions over the area is crucial for the climate model improvements. The study has revealed many key discrepancies between observations and WRF-CAM5; for example, the lower hygroscopicity κ in the simulations, the lack of aerosol loss processes in the model that may have led to an erroneous aging process and/or number concentrations, and the rainfall tendencies (i.e., the drizzle problem). These points are important for the future aerosol/cloud model developments, and hence the study conveys important messages. Figures and tables present simulated and/or observed results well, which get explained thoroughly in the main text. The structure of the paper and the presentation of the results can be somewhat improved before the publication, as it may be confusing in the current format. While this is understandable given the number of different variables presented from many field campaigns and instruments, I suggest the re-organization of the paper structure so that a few most important messages stand out, rather than keeping them as one of many findings, which I believe will convey the messages more easily to the readers.

**Specific comments**

General structure: Given the amount of information in the paper, I believe the clear structure of the paper is crucial for readers to understand the content as much as possible. Currently the main section is structured as follows;

3. Results

3.1 Free troposphere smoke

3.1.1 Physical properties

3.1.2 Aging processes in the free troposphere, WRF-CAM5 vs. ORACLES

3.2 Boundary layer performance

3.2.1 Boundary layer smoke composition

3.2.2 Boundary layer size distributions

3.2.3 Hygroscopicity in LASIC

3.2.4 Smoke entrainment, removal, and rain at Ascension Island

3.2.5 Aerosol activation and turbulence

For example, before starting 3.1 (right after "3. Results"), the authors can explain that the results are presented firstly for the free troposphere and secondly for the boundary layer, and why the two are presented separately. This will give readers an idea of the overall structure and the reasons for the separation. The same thing can be done right after "3.1" and "3.2", so that readers understand why the subsection is split into multiple subsubsections, and what contents are expected within the subsection. This also applies to Section 2 (Methods) as well; the authors can add a few sentences between "2. Methods" and "2.1 Observation systems" on what each subsection contains and why the section is split into five subsections.

- We've added a brief intro after the section 2 header to outline and organize the methods, after section 3 header to describe the results, as well as the split between the FT and BL analyses. We've also added an intro to the FT section which summarizes what will be analyzed in section 3.1, and a similar overview at the end of the 3.2 text before section 3.2.1 begins.

The subsection title for 3.1 and 3.2 should be comparable, for instance, "Free troposphere" and "(Marine) Boundary layer". Once these are defined in the subsection titles, then the subsubsection titles do not need to include "free troposphere" or "boundary layer" again (as it's obvious). Ideally, 3.1.1 and 3.2.1 have comparable titles and contents, for example "Physical properties" for both 3.1.1 and 3.2.1, so that each subsection has the same/similar subsubsections and similarly structured. While this may be some work and re-organization of the contents, I believe the paper will convey the messages better to readers. It would be even better if each subsubsection has a clear message or finding, which gets summarized at the end in Conclusions (and also in Abstract).

- Both sections 3.1 and 3.2 have been lightly reorganized to make the presentation more consistent. 3.1 and 3.2 (FT and BL) now both begin with discussion of extensive properties (measures of smoke amount) and size properties/distributions. They next discuss chemical composition and hygroscopicity. The last sections of the FT and BL results (aging in the FT, and removal, turbulence, and cloud activation in the BL) are unique to each regime. We've also clarified the findings from each section in the conclusion, in the new ordering.

Abstract: Related to my general comments above, the abstract seems to include too much detailed information on many topics, but it should be a concise summary of a few key findings of the paper. In addition, in the current format, readers may get the impression that aerosol size and composition in the free troposphere are the only variables that the model was able to simulate well, and the rest was not well simulated/reproduced, though this is not the main point of the paper. I believe summarizing a few key discrepancies and their cause in brief words will give a better idea of the paper content.

- We've rewritten the abstract to include fewer details and explanations and focus on the main findings and their primary reason.

Figure 2: The red crosses in the plots are not explained, but I believe these are outliers? Please add this information to the figure caption.

- We've added this text to the caption for figure 2:
- ", and small red crosses are outliers (greater than 1.5 times the interquartile range beyond the box)."

Line 418: Should this be "above" 3 μm rather than "below", if the topic is the lack of coarse-mode particles?

- This was intended to say that we have no evidence of a coarse mode, but that the particle sampling data from the PCASP is limited to measuring 3 μm particles at the largest. Larger particles could exist above this, but it is very unlikely based on human flight reports and the otherwise lack of even a partial coarse dust mode in observations. This caveat added more confusion than necessary to make this point, so we've trimmed the sentence in question to read:
- "…there is not a substantial volume of coarse particles such as mineral dust or sea spray."

Line 588: "excluding the clean period" – why was it excluded? From the figure, the overprediction is also seen during the clean period.

- The clean period was excluded to avoid biasing the mean when comparing *smoky* BL air. We've added that the overprediction during the clean period is far greater, roughly 1,000% higher.

Conclusions: I suggest that the authors list the key findings either in bullet points or in numbers, and make them concise. The detailed numbers do not need to be repeated here for all the items, except for a few key important numbers.

- We've trimmed down the detailed numbers from the conclusion, as well as reorganized it to match the new organizational structure of the body of the paper.

Figures 2g-h, 4b-I, 6f-g: Some of these figures are not referred to in the manuscript, or their

contents are discussed but they are not exactly referenced/referred to in the manuscript.

- We have updated section 3.1.1 to address Figs. 2g-h, and sections 3.1.2 and 3.2.2 to explicitly refer to the pie charts in comparing FT and BL compositions. Figs. 6f-g are already referenced in section 3.2.3 about boundary layer heights and inversion strength.

Figure A3: Is this figure referred to in the manuscript?

- We've noted that this figure provides some baseline for background CO level estimation in section 3.2.3, related to estimates of removal at ASI:
- "A higher assumed background CO of 60 ppb—as seen in a fire-off run of WRF-CAM5 over this same period (Fig. A3)—would only amplify this discrepancy."

Technical corrections

Abbreviations: Throughout the paper, there are a few words that are defined earlier (e.g., ASI, FT,

SEA, MBL) but the full words are used again afterwards, or defined twice. I suggest that the authors make sure abbreviations are defined only once, and they're used throughout the paper after

- We have corrected usage of MBL, FT, SEA, and ASI for consistency. We've elected to still write out these acronyms in section headers so a reader can skim the paper structure more easily without needing to parse acronyms.

the definitions.

Line 73: "Intergovernmental", not "International".

- Corrected

Line 238: Add "(2022)" after "Barrett et al."

- Corrected

Line 268: Missing the reference for the WRF-Chem model.

- Added reference to Skamarock et al., (2008).

Line 272: Delete "Niemand et al., " in the bracket, as only the year is necessary here.

- Corrected

Line 427: I believe this is a typo, the "µ" at the end needs to be deleted.

- Corrected

Line 516: This should be "downward" without an "s".

- Corrected

Line 563: I believe it should be either "…, which is generally" or "…, (which is) a difference that is generally"

- Corrected to read "…a large difference in particle composition between the FT and the boundary layer (Fig. 4) that is generally not captured in WRF-CAM5"

Line 595: It should be either ", which corresponds to…" (with a comma) or "that corresponds to…"

- Corrected to read "…that corresponds to…"

Line 598: Remove "which"

- Removed

Line 598: "smoke-free" instead of "smoky free"

- "smoky" describes the FT for this sentence. Using the FT acronym, the meaning should be more clear.

Line 635: Add "κ" between "than" and "in"

- Added

Line 671: Add a period after "periods" to end the sentence.

- Added

Line 673: "periods" instead of "period"

- Corrected

Line 721: I think it should be "provides" instead of "providing", assuming that the subject is "This" in the beginning. Or, it should be "…, while providing…".

- Corrected

Figure 9 caption: Add "and" between "each campaign" and "WRF-CAM5" (6th line).

- Corrected

Line 777: Add "to" after "due".

- Corrected

Line 786: Add "MBL" before "compared to".

- Corrected

Line 797: The beginning of this sentence needs to be corrected – I believe it is "The fact that"?

- Corrected

Line 817: "are often when" should be "often correspond to the time period when" or something similar.

- Rewrote

Line 834: Replace "measure" with either "compare" or "evaluate".

- Changed to "compare"

Line 836: This sentence is more appropriate in Introduction, not in Conclusions.

- Deleted the sentence here referring to WRF-CAM5 performance in previous studies.

Line 886: Add a dot after "comparison"

- Added

**Additional changes:**

During the review process, the authors realized some minor changes and clarifications in the manuscript were appropriate, which are noted below.

- Figure 9a-b: the PDFs on their respective axes updated. Previously, the PDFs were using a kernel averaging technique to plot a smooth curve. That may have given the incorrect impression that the data could be extrapolated beyond the datapoints shown in the scatter plot. The PDFs have been updated to show simple bins/frequencies matching the data.
- The discussion related to TKE has been corrected. The quantity calculated from the standard deviation of the vertical winds, as well as that in the model output, represents not TKE (units of $m^2\,s^{-2}$), but instead a characteristic turbulent updraft *velocity* $w^*$ (units $m\,s^{-1}$). This is a textual error, not one of calculation or interpretation, and the conclusions about turbulence still hold. The units of $w^*$ have been corrected through the text and in figure 9e-f, as well as references to TKE. We have clarified lines 213-217 to read as follows, incorporating the referee's comment to clarify the 0.79 factor:
"Vertical turbulence was approximated using vertical wind measurements from a high-resolution anemometer (Morales & Nenes, 2010). This calculation fitted a Gaussian curve to the updraft spectrum integrated over 1024 samples at 20Hz. The characteristic turbulent updraft velocity ($m\,s^{-1}$), proportional to the root of turbulent kinetic energy ($TKE^{1/2}$), was taken as 0.79*σ, where σ is the standard deviation of that Gaussian curve. The factor of 0.79 also comes from the derivation in Morales & Nenes (2010). This quantity is also output directly from WRF-CAM5, where it is used with the grid-scale updraft speed to construct a Gaussian updraft spectrum that is then used to calculate activation. Both characteristic updrafts are selected in the vertical range of 100-700m that contained most flat BL flight legs."
- We have added some clarifying comments in section 3.2.2 about $SO_2$ co-emitted with smoke, and that a modeled bias may be contributing to the discrepancy in BL sulfate concentrations. $SO_2$ is very inefficient at condensation without aqueous oxidation, which is much more prevalent in the boundary layer. A deeper analysis of this is beyond the scope of this work.